# Serotonin modulates asymmetric learning from reward and punishment in healthy human volunteers

Jochen Michely [1,2,3,4], Eran Eldar [5], Alon Erdman [5], Ingrid M. Martin[4,6] & Raymond J. Dolan [3,4]

Instrumental learning is driven by a history of outcome success and failure. Here, we examined the impact of serotonin on learning from positive and negative outcomes. Healthy human volunteers were assessed twice, once after acute (single-dose), and once after prolonged (week-long) daily administration of the SSRI citalopram or placebo. Using computational modelling, we show that prolonged boosting of serotonin enhances learning from punishment and reduces learning from reward. This valence-dependent learning asymmetry increases subjects' tendency to avoid actions as a function of cumulative failure without leading to detrimental, or advantageous, outcomes. By contrast, no significant modulation of learning was observed following acute SSRI administration. However, differences between the effects of acute and prolonged administration were not significant. Overall, these findings may help explain how serotonergic agents impact on mood disorders.

[1] Department of Psychiatry and Neurosciences, Charité – Universitätsmedizin Berlin, Corporate Member of Freie Universität Berlin and Humboldt-Universität zu Berlin, Berlin, Germany. [2] Berlin Institute of Health at Charité – Universitätsmedizin Berlin, BIH Charité Clinician Scientist Program, Berlin, Germany. [3] Max Planck UCL Centre for Computational Psychiatry and Ageing Research, University College London, London, UK. [4] Wellcome Centre for Human Neuroimaging, University College London, London, UK. [5] Psychology and Cognitive Sciences Departments, Hebrew University of Jerusalem, Jerusalem, Israel. [6] Institute of Cognitive Neuroscience, University College London, London, UK. ✉email: jochen.michely@charite.de

To maximize reward, and minimize punishment, agents need to learn from a history of past success and failure[1]. Evidence suggests that, rather than being represented on a continuous scale, reward and punishment may represent distinct categorical events[2,3]. This is corroborated by findings that exposure to reward and punishment engage distinct brain networks[4–7]. This triggers distinct types of behaviour, such as approach and invigoration for reward, and avoidance and inhibition for punishment[8,9].

Numerous studies report an asymmetric impact of positive and negative outcomes during human instrumental learning[10–12]. Interestingly, such valence-dependent learning asymmetries are characterised by remarkable flexibility, for example an adjustment to environmental volatility[13] or contextual information[14]. More-over, learning asymmetries are linked to interindividual variability in brain structure and function[7,15], and are thought to play a role in the emergence of mood disorders, often characterised in terms of aberrant processing of reward and punishment[16,17].

Previous research shows that the neuromodulators dopamine and serotonin play a key role in modulating reward and pun-ishment learning. Whilst there is ample evidence for a role of dopamine in learning from reward[18–21], the evidence in relation to serotonin is less clear. Some studies indicate a specific role in punishment learning[22–26], while others report that serotonin impacts learning from both reward and punishment[27,28].

A key challenge studying the human serotonergic system is the manipulation of central serotonin levels. In previous studies, serotonin has often been modulated via a dietary depletion of tryptophan, a precursor of serotonin[29]. However, conclusive evidence supporting the effectiveness and specificity of this dietary manipulation in humans is lacking[30], but cf.[31].

Arguably, a more specific method involves a pharmacological enhancement of central serotonin via the use of selective ser-otonin reuptake inhibitors (SSRIs). To date, SSRI studies of learning asymmetries have been mainly limited to single dose administration or assessment of clinical populations, which makes interpretation problematic[26,27,32]. Human and non-human animal data suggest a pharmacological modulation of serotonin can impact brain function at different timescales, with distinct effects of single dose, one-day intervention, and pro-longed, repeated administration over several days[33–37]. This accords with the fact that SSRIs reach steady-state peak plasma levels only after a prolonged treatment spanning multiple days[38]. This renders it likely that prolonged and repeated administration of serotonergic drugs is necessary to impact on behaviour, such as inducing a substantial modulation of learning processes[28,39].

Based on these considerations, we examined the impact of an extended exposure to a serotonergic treatment on human reinfor-cement learning. Healthy human volunteers were assessed twice, once after single dose, and once after repeated daily administration of either 20 mg of the SSRI citalopram, or placebo, across seven con-secutive days. Subjects performed a task specifically tailored to study an asymmetry in learning from reward and punishment[7]. We used computational modelling to examine fine-grained characteristics of learning over time. We show prolonged SSRI treatment exerts an asymmetric impact on reinforcement learning, reducing learning from reward and enhancing learning from punishment. We discuss the implications of these findings with respect to potential mode of action of serotonergic treatment in the context of mood disorders.

## Results

We administered sixty-six healthy volunteers either a daily oral dose of the SSRI citalopram (20 mg) or placebo, across seven consecutive days. Subjects performed two experimental sessions, once after administration of a single dose on day 1 and once, after exposure to repeated daily administration, on day 7 (Fig. 1b).

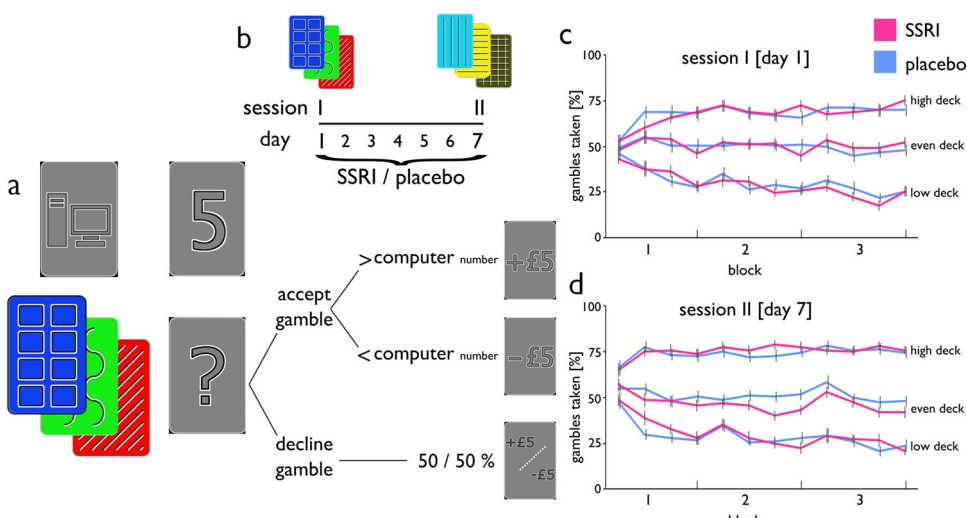

**Fig. 1 Experimental task, pharmacological procedure, and learning performance. a** Experimental design: On each trial, participants were presented with one of three possible decks and a number between 1 and 9 drawn by the computer. If participants decided to gamble, they won £5 if the number they drew was higher than the computer drawn number, and lost £5 if the number was lower. Participants were only informed whether they won or lost the gamble, not which number they drew. Participants had to learn by trial and error how likely gambles were to succeed with each of the three decks. One deck contained a uniform distribution of numbers between 1 and 9 (even deck), one deck contained more 1's (low deck), making gambles 30% less likely to succeed, and one deck contained more 9's (high deck), making gambles 30% more likely to succeed. Opting to decline the gamble resulted in a 50% probability of win/loss, regardless of which number was drawn by the computer. **b** Pharmacological procedure: Subjects were randomly allocated to take a daily dose of 20 mg citalopram or placebo for seven days and participated in two sessions: session I took place on day 1 after single administration, session II took place on day 7, at a time when citalopram reaches steady-state plasma levels. Subjects played an identical game on both sessions, but with two independent sets of three decks, where colour order and colour-associated win probability randomly varied across participants. **c**, **d** Learning performance: Gambles taken with each deck as a function of time. Percentages were computed separately for each set of 15 contiguous trials (4 sets/60 trials per block), for session I (**c**), and session II (**d**), respectively. Error bars indicate SEM.

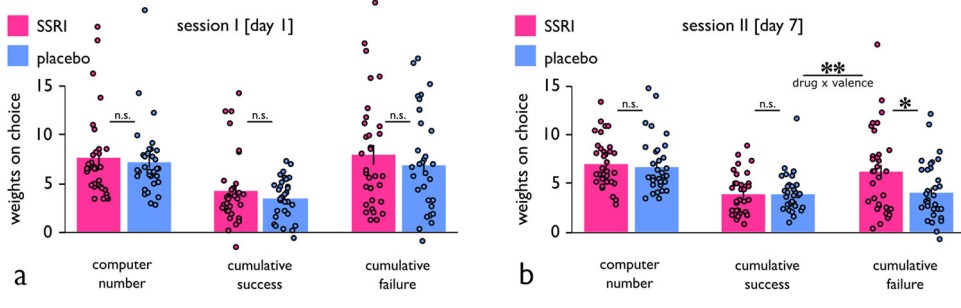

**Fig. 2 Results of trial-by-trial logistic regression model.** Fitting a logistic regression model to subjects' decisions showed that participants gambled more against lower computer numbers, with no drug differences on session I (**a**), and session II (**b**), respectively. Additionally, subjects, over the course of a session, gambled more with increasing success with each deck, and gambled less with increasing failure with each deck. On session I, impact of cumulative success and failure was unaffected by treatment (**a**). On session II, however, SSRIs enhanced the impact of failure but not wins (**b**), indicating an asymmetric drug effect on reward and punishment. **\*\****p* < 0.01, **\****p* < 0.05, n.s. = not significant (*p* > 0.05). Error bars indicate SEM.

During each session, subjects performed a modified version of a gambling card game (Fig. 1a;[7]), where the goal was to maximize monetary wins and minimize monetary losses.

In brief, on each trial participants were presented with a number between 1 and 9 as drawn by a computer. Subjects could gamble that the number they were about to draw would be higher than this computer drawn number. Critically, participants played with one of three possible decks on each trial, where the decks differed in how likely gambles were to succeed. One deck contained a uniform distribution of numbers between 1 and 9 (even deck), one deck contained more 1's (low deck, gambles 30% less likely to be successful), and one deck contained more 9's (high deck, gambles 30% more likely to be successful).

Subjects were informed that an unsuccessful gamble would result in a loss (−£5), and a successful gamble would result in a win (+£5). Subjects learnt through trial and error about each of the decks' success likelihood. Alternatively, subjects could decline a gamble and instead opt for a fixed 50% known probability of winning or losing, respectively. After a decision to decline a gamble, the outcome was not shown to participants.

On a second session, the game was identical, and the only difference being that subjects played with three novel decks, indicated by different colours, where colour order and colour-associated win probability was randomly varied across participants (Fig. 1b). Subjects had to learn about these decks anew as they were unrelated to the ones from the first session.

The experiment was designed such that the computer numbers changed over time to ensure subjects gambled on approximately 50% of trials across all decks (cf. Methods). Indeed, this adaptation worked, and overall proportion of accepted gambles did not differ between drug groups (session I: SSRI 49.0%, placebo 49.3%, $t_{65} = 0.1$, $p = 0.846$; session 2: SSRI 50.6%, placebo 51.2%, $t_{65} = 0.4$, $p = 0.641$). Thus, evidence of learning manifested in how the rate of accepted gambles differed between decks, and in an observation that this difference grew over the course of the experiment, i.e., from 1st to 3rd block (Fig. 1c/d; session I: low vs. even, $t_{65} = 3.2$, $p = 0.0016$; low vs. high, $t_{65} = 6.1$, $p = 6.0e−8$; even vs. high, $t_{65} = 3.0$, $p = 0.003$; session II: low vs. even, $t_{65} = 2.9$, $p = 0.003$; low vs. high, $t_{65} = 6.1$, $p = 6.2e−8$; even vs. high, $t_{65} = 2.7$, $p = 0.007$). There was no significant difference between the groups in this respect (session I: low vs. even, SSRI vs. placebo: $t_{64} = 1.4$, $p = 0.159$; low vs. high, SSRI vs. placebo: $t_{64} = 1.0$, $p = 0.291$; even vs. high, SSRI vs. placebo: $t_{64} = −0.2$, $p = 0.783$; session II: low vs. even, SSRI vs. placebo: $t_{64} = 0.06$, $p = 0.949$; low vs. high, SSRI vs. placebo: $t_{64} = 0.6$, $p = 0.491$; even vs. high, SSRI vs. placebo: $t_{64} = 0.6$, $p = 0.549$). Overall, this demonstrates that subjects learned to dissociate decks, as their

willingness to gamble differed depending on each deck's win likelihood as a function of time, and this effect was not modulated by the drug.

Next, we used a trial-by-trial logistic regression approach (cf. Methods) to assess whether subjects' decisions to gamble were dependent upon the computer number and previous receipt of positive, or negative, outcomes over time. First, we found that subjects, over both sessions, gambled more against lower computer numbers (session I: $t_{64} = 14.7$, $p = 3.0e−22$; session II: $t_{65} = 20.9$, $p = 1.3e−30$; Fig. 2), with no evidence for a difference between drug groups (drug: $F_{1,63} = 0.4$, $p = 0.505$; drug x session: $F_{1,63} = 0.06$, $p = 0.805$).

Second, participants, over the course of each session, gambled more with decks with which they had experienced more success (session I: $t_{64} = 10.3$, $p = 3.0e−15$; session II: $t_{65} = 15.7$, $p = 8.2e−24$) and less failure (session I: $t_{64} = 10.2$, $p = 4.3e−15$; session II: $t_{65} = 11.0$, $p = 1.6e−16$). This result indicates subjects successfully learned about the decks from the outcomes of their gambles. When assessing data across both sessions, the pharmacological effect on gambling preferences as a function of outcome type was not statistically significant (drug: $F_{1,63} = 1.7$, $p = 0.194$, drug x valence: $F_{1,63} = 2.6$, $p = 0.108$, drug x session x valence: $F_{1,63} = 2.4$, $p = 0.124$). However, analysing both sessions separately, effects were similar across drug groups for cumulative success and failure on session I (drug x valence: $F_{1,63} = 0.3$, $p = 0.844$; drug: $F_{1,63} = 0.9$, $p = 0.345$; Fig. 2a), whereas on session II we found evidence for an asymmetric impact of success and failure outcomes, as a function of treatment (drug x valence, $F_{1,64} = 10.5$, $p = 0.0018$; drug: $F_{1,64} = 2.4$, $p = 0.126$; Fig. 2b), attributable to an enhanced impact of failure ($t_{64} = 2.3$, $p = 0.024$) but not of success ($t_{64} = 0.1$, $p = 0.892$), in SSRI treated as compared to placebo subjects. This differential pattern suggests that SSRI treatment increased an impact of negative outcomes, enhancing a gamble avoidance tendency in response to failure.

Next, we used computational modelling (cf. Methods) to assess the precise learning mechanism underlying the asymmetric effects of success and failure evident in the regression analysis. Replicating results from an earlier study using an identical cognitive task[7], model comparison showed task behaviour was best explained by a model that accounted for an asymmetry in learning from the two outcome types. Specifically, the best-fitting model included two different learning rates, one for wins ($\eta^+$), and one for losses ($\eta^-$), where these determine the degree to which an outcome type impacts on subsequent expectations (model 6: '*adjusted & asymmetric Q-learning*'; cf. Supplementary Table 1 for iBIC scores). These expectations, in combination with the numbers drawn by the computer, shape whether gambles are

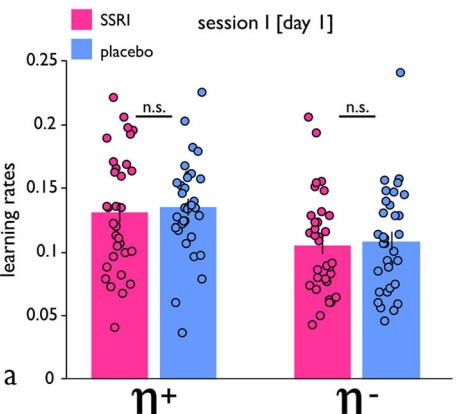
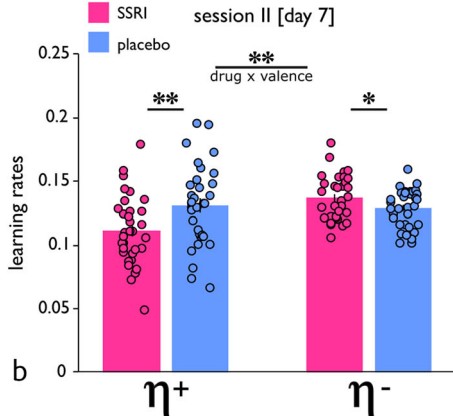

**Fig. 3 Learning asymmetry and its serotonergic modulation. a** On session I, computational modelling showed learning from reward ($\eta^+$) and learning from punishment ($\eta^-$) was unaffected by drug treatment. **b** On session II, SSRI treatment had a significant effect on learning asymmetries, such that it reduced learning from reward ($\eta^+$), and enhanced learning from punishment ($\eta^-$). Note that data simulation showed that an 'optimal' learning rate for the task, maximizing net reward gained, was in the range of ≈0.3–0.6 (Supplementary Fig. 2). **\*\***$p < 0.01$, **\***$p < 0.05$, n.s. = not significant ($p > 0.05$). Error bars indicate SEM.

likely to be taken or declined. The predictive accuracy of the model (absolute fit), i.e., the proportion of subjects' choices to which the model gives a likelihood greater than 50% (percent correct), was 87.71% for session I, and 87.92% for session II (Supplementary Fig. 1).

When assessing computational parameters across data from both sessions, we found a significant asymmetric effect of SSRIs on learning rates (drug x valence: $F_{1,62} = 4.1$, $p = 0.046$; drug: $F_{1,62} = 0.8$, $p = 0.365$), but no significant three-way interaction (drug x valence x session: $F_{1,62} = 1.0$, $p = 0.305$, controlling for an overall gambling bias, Supplementary Fig. 3). Follow-up tests revealed that, on session I, there was no evidence for computational parameters governing the rate of learning from reward and punishment being different between treatment groups (drug x valence: $F_{1,64} = 0.007$, $p = 0.933$; drug: $F_{1,64} = 0.3$, $p = 0.553$; Fig. 3a). However, by session II, a significant serotonergic impact on learning asymmetry was evident (drug x valence: $F_{1,64} = 8.2$, $p = 0.006$; drug: $F_{1,64} = 3.2$, $p = 0.075$; Fig. 3b), such that in SSRI, as compared to placebo subjects, learning from reward was reduced ($t_{64} = 2.7$, $p = 0.008$) while learning from punishment was enhanced ($t_{64} = 2.0$, $p = 0.041$).

Overall, these results indicate that a prolonged regimen of SSRI treatment resulted in a modulation of learning asymmetries. Importantly, there were no between-group differences for the remaining model parameters (Supplementary Fig. 3). With regards to the impact of a single SSRI dose, on the one hand there was no significant impact following a single dose, while on the other the impact following a single dose did not significantly differ from the impact following prolonged SSRI treatment.

Additionally, an asymmetric effect of cumulative success and failure on gambling, as derived from the logistic regression, correlated significantly with an asymmetry in learning, as derived from the computational reinforcement learning model, in both sessions for both drug groups (session I, placebo: $r = 0.873$, $p = 7.1e{-}11$, session I, SSRI: $r = 0.819$, $p = 5.7e{-}9$; session II, placebo: $r = 0.841$, $p = 8.7e{-}10$; session II, SSRI: $r = 0.737$, $p = 9.8e{-}7$; Fig. 4).

The benefit of having both analyses is that the model-based analysis is more sensitive, albeit at a cost of greater flexibility in fitting the data. Specifically, reinforcement learning modelling can mimic our regression analysis by fitting the data with very low learning rates, thus weighting outcomes almost equally. However, by fitting the data with higher learning rates, it can also place substantially greater weight on recent outcomes. We additionally illustrate the correspondence between these two measures

(regression and reinforcement learning modelling) in simulations with a wide range of parameter settings. Briefly, we simulated artificial data from five models, in which we randomly varied positive and negative learning rates independently across agents. Next, we ran logistic regression analyses on the artificial data and computed correlations between an asymmetric effect of cumulative success and failure on gambling (regression) and an asymmetry in learning (computational model). Here, we found a highly significant relationship across all simulated data sets (r ranging between 0.82–0.87, all $p < 2.6e{-}17$), providing further evidence for the relationship between these two measures. Overall, these analyses jointly indicate asymmetric learning from positive and negative outcomes related to an altered gambling preference and was influenced by prolonged serotonergic intervention.

Note in addition we tested a model with differing sensitivity to outcome valence (positive and negative, respectively), and a model that modulates both a sensitivity to distinct outcomes and learns differently from these distinct outcomes. Across both sessions, the models provided a worse fit than a model that learns asymmetrically from distinct outcomes (Supplementary Table 1). Although the latter model was clearly not the best-fitting model, we used it for a joint test of drug effects on both outcome sensitivity and learning parameters from session II. Here we found that a sensitivity to outcomes did not differ between drug groups ($t_{64} = 0.5$, $p = 0.582$), but that asymmetric learning was modulated by SSRIs (drug x valence: $F_{1,64} = 10.4$, $p = 0.0019$), with reduced learning from reward ($t_{64} = 2.9$, $p = 0.004$), and an increased learning from punishment ($t_{64} = 2.1$, $p = 0.032$) following serotonergic intervention, mirroring the drug effects on parameters of our winning model.

Note that, across all subjects and sessions, we found no significant difference between positive and negative learning rates ($F_{1,65} = 2.1$, $p = 0.151$). However, in placebo subjects alone, we found a statistical trend for greater learning from positive as compared to negative outcomes ($F_{1,32} = 3.7$, $p = 0.063$). This is in line with recent work showing a learning asymmetry towards greater updating from positive information healthy individuals, without pharmacological intervention[15]. We found no difference between drug groups in net reward gained, a key measure for task performance (session I: $t_{64} = 1.2$, $p = 0.201$; session II: $t_{64} = 0.4$, $p = 0.650$). Thus, we found no evidence that changes in asymmetric learning were detrimental, or advantageous, for task performance.

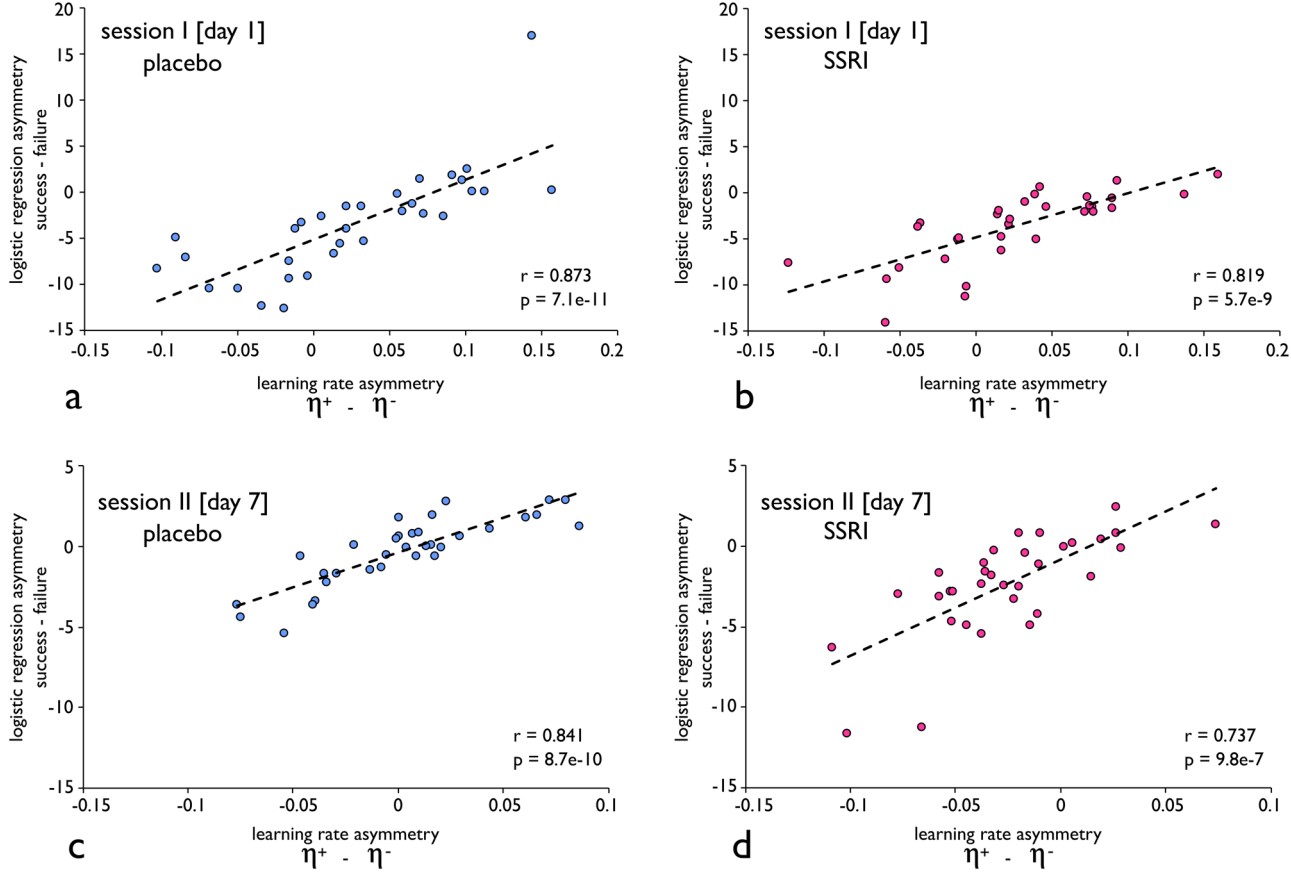

**Fig. 4 Asymmetric effects of reward and punishment.** An asymmetric effect of cumulative success and failure on gambling, as derived from the logistic regression, significantly correlated with an asymmetry in learning, as derived from our computational model, in both sessions for both drug groups. **a** Session I, placebo. **b** Session I, SSRI. **c** Session II, placebo. **d** Session II, SSRI.

We performed several analyses to assess the validity of our computational modelling approach. First, we generated simulated data based upon model parameters derived from fitting to real data. This 'posterior predictive check' confirmed that the model captured core features of the real data (Supplementary Fig. 4). Additionally, we simulated sets of choices from artificial agents with specific sets of parameters ('ground truth') and then fitted models to those choices to recover the values of the parameters ('recovered parameters'). To ensure the results of the parameter recovery test were applicable to the analysis of the real data, we selected the ground truth parameters such that they covered the empirical range. This analysis revealed that individual parameter estimates could be accurately recovered (Supplementary Figs. 5/6). Lastly, we validated our model comparison procedure by generating simulated data using each model and applying our model comparison procedure to identify the model that generated each dataset (Supplementary Fig. 7).

## Discussion

Here, we show that boosting central serotonin by means of week-long SSRI administration enhanced learning from punishment and reduces learning from reward. This SSRI-induced learning asymmetry increased subjects' tendency to avoid actions as a function of cumulative failure.

Serotonin is an evolutionary conserved neurotransmitter though its precise effects on cognition has evaded a definite mechanistic understanding[40,41]. One influential proposal is that serotonin plays a specific role in processing aversive outcomes[42]. Indeed, several studies in humans show that serotonin is involved in punishment learning[22–26], but other studies suggest that it impacts learning from both positive and negative outcomes[27,28].

In our study, we replicate findings from a previous study that used an identical learning task[7], showing behaviour is best explained by an asymmetry in reward and punishment learning. A strength of our task is that learning from reward and punishment are each assessed via their naturally associated go (i.e., accept gamble) or no-go (i.e., reject gamble) Pavlovian responses[9]. Additionally, reward and punishment are administered within the same block, in an interleaved manner, thereby competing for a subject's learning resources[43]. Unfortunately, in this study, we were not in a position to acquire neural data. In light of previous studies on interindividual variability in human learning asymmetries, it is tempting to speculate that serotonergic agents may act preferentially in the striatum and prefrontal cortex to alter the relative degree of impact from positive and negative outcomes[7,15,44].

The effects of a serotonergic manipulation we highlight require a temporally extended treatment in order to evolve. This accords with human and non-human animal studies showing that only a prolonged modulation of serotonin induces a substantial impact[33,36,45,46], particularly with respect to learning[28,35]. The fact that changes in learning emerge after an extended intervention may reflect two processes, or a synergism of both. First, citalopram reaches steady-state plasma levels after seven days[38], and a single dose administration is unlikely to suffice for induction of a substantial modulation of learning. Second, plasticity that may underlie this modulation, such as s neurogenesis, synaptogenesis, or changes in BDNF levels, require days or weeks to emerge[47].

A learning asymmetry, involving a greater impact of losses than wins, can lead to increased avoidance relative to approach behaviour. This can result in an aversion to risk-taking and action over time, and potentially maladaptive risk-avoidant behaviour[48]. However, studies specifically assessing the impact of serotonin on human risk-taking have, thus far, proven inconclusive[29,49–51]. Notably, in these studies, decision variables are typically not learned by trial and error but are instead presented to subjects explicitly, which contrasts with the learning design utilised in the current study. Thus, the relationship between a serotonergic effect on asymmetric learning and the development of risk tendencies remains a question for pursuit in future studies.

Aberrant processing of reward and punishment is assumed to play a role in the emergence of mood disorders[16,17]. Although SSRIs constitute a first-line pharmacological intervention in mood disorders[52], the cognitive and computational mechanisms that underpin treatment effectiveness remain an unresolved issue[53]. We suggest the asymmetric effects we highlight can help explain the clinical impact of SSRIs. Specifically, recent lab-based[54–57] and real-world studies[58,59] show that outcome prediction errors, or positive and negative surprise, strongly impact on subjects' emotional state. Here, an agent's mood depends not only on how well things are going in general, but whether things are better, or worse, than expected. Our results indicate that a serotonergic intervention can, in principle, influence the affective impact of reinforcement, by lowering positive expectations through slowing reward learning, thereby giving rise to more positive surprise, and minimizing the impact of aversive outcomes by enhancing negative expectations as a function of increased punishment learning. Overall, a consequence is an increase in positive as compared to negative surprise, an effect that may contribute to a gradual emergence of better mood[60].

A related line of work shows that healthy individuals learn more from positive, relative to negative, information leading to an 'optimism bias'[61,62], where the latter is lacking in depressed patients[63,64]. This might seem to suggest that an increased learning from positive information acts to protect against depression. It is worth highlighting, however, that these studies typically assay updating expectations about oneself given information about an average person. This type of assay is important, but it involves an additional critical factor—the degree to which subjects accept that information about the average applies to themselves. There are a range of reasons to discount the relevance of average statistics, and such discounting can be applied asymmetrically to positive and negative information. We propose this factor, rather than differences in learning per se, might explain depression-related individual differences in optimism bias. In contrast, our approach strives to assess a more basic process of learning from reward and punishment. For this we designed our study such that the feedback from which subjects learn is unequivocal, as is typically applied in the basic reinforcement learning literature[7,10,11].

Overall, we consider that results, alongside its putative impact on changes in mood, do not contradict findings on optimism bias in depression. On the contrary, both processes can be expected to contribute to an emergence of better mood[60]. Here, a limitation of our study is its restriction to non-depressed healthy individuals. Moreover, week-long SSRI treatment does not typically result in a meaningful mood improvement in a clinically depressed population[65]. Furthermore, our task did not contain a concurrent mood measurement. Ultimately, testing both self-referential and basic reinforcement learning, alongside tracking of subjective changes in mood in clinical cohorts, will be an important next step for examining a temporal evolution of changes in learning and the emergence of clinical effects over time.

Although our data suggests an emergence of serotonergic effects after a temporally extended intervention, it is of note that a three-way interaction (drug x session x valence) was not significant. Thus, we tentatively conclude that prolonged treatment induced a learning asymmetry, but the interpretation of this needs to be tempered by the fact that there was no difference between prolonged (week-long, on day 7) as compared to acute (single-dose, day 1) treatment. To unravel the precise trajectory of any such effect, future studies should ideally include a pre-drug testing session as well as multiple sessions over several weeks of treatment.

In summary, we show that week-long SSRI treatment reduces reward and enhances punishment learning. This learning asymmetry can, in theory, result in lowered positive and enhanced negative expectations, and consequentially, to more rewarding, and less disappointing experience. We suggest this modulation of computations that guide reinforcement learning may contribute to a known serotonergic impact on mood.

## Methods

**Subjects.** Sixty-six healthy volunteers (mean age: 24.7 ± 3.9; range 20–38 years; 40 females; Supplementary Table 2) participated in this double-blind, placebo-controlled study. All subjects underwent an electrocardiogram to exclude QT interval prolongation and a thorough medical screening interview to exclude any neurological or psychiatric disorder, any other medical condition, or medication intake. Subjects were reimbursed for their time. Additionally, subjects were informed that, at the end of the experiment, one trial was randomly selected, and the outcome of that trial was added to the overall payment. Thus, performance was incentivised as choosing good gambles resulted in a higher probability of earning additional monetary reward. Data from different tasks of the same participants are reported elsewhere[66,67]. The experimental protocol was approved by the University College London (UCL) local research ethics committee, with informed consent obtained from all participants.

**Pharmacological procedure.** Participants were randomly allocated to receive a daily oral dose of the SSRI citalopram (20 mg) or placebo, over a period of seven consecutive days. All subjects performed two laboratory testing sessions. Session I was on day 1 of treatment, 3 h after single dose administration, as citalopram reaches its highest plasma levels after this interval[68]. On the following days, subjects were asked to take their daily medication dose at a similar time of day, either at home or at the study location. Session II was on day 7 of treatment, a time when citalopram is known to reach steady-state plasma levels[38], with the tablet being taken 3 h before test. Thus, subjects were assessed twice, once after single-dose, and once after week-long administration of the drug. This repeated-measures study design enabled (i) an assessment of a pharmacological effect overall, as both sessions were performed under the influence of the drug, and (ii) an assessment of putative differences between acute (single-dose) and prolonged (week-long) treatment.

**Affective state questionnaires.** To examine putative effects of the drug on subjective affective states over the course of the study, participants completed the Beck's Depression Inventory (BDI-II,[69]), Snaith-Hamilton Pleasure Scale (SHAPS,[70]), State-Trait Anxiety Inventory (STAI,[71]), and the Positive and Negative Affective Scale (PANAS,[72]) on two different occasions: (i) pre-drug, day 1; (ii) peak drug, day 7.

**Experimental task.** To examine differences in learning from success and failure, we used a modified version of a gambling card game[7], in which subjects' goal was to maximize monetary wins and minimize monetary losses.

The game consisted of 180 trials, divided into three 60-trial blocks. On each trial (Fig. 1a), subjects were shown with which one of three possible decks (each designated by distinct colour and pattern) they will be playing. After a short interval (2 to 5 s, uniformly distributed), the computer drew a number between 1 and 9, and participants had up to 2.5 s to choose whether they wanted to gamble that the number, which they are about to draw, will be higher than the computer drawn number. If participants chose to gamble, they won £5 if the number that they drew was higher than the computer's number, and they lost £5 if it was lower (as well as in half of the trials in which the numbers were equal). If subjects opted to decline the gamble, they won/lost with a fixed 50% known probability. On such trials, the outcome was not shown to participants. Not making any choice always resulted in a loss. Feedback was provided 700 ms following each choice and consisted of a '+£5' (win), '−£5' (loss), or '+£5 / −£5' (win or loss, 50% probability) visual symbol. The drawn number was not shown. Subjects were told that each of the three decks contained a different proportion of high and low

numbers, and they could learn by trial and error about each of the decks' likelihood of success.

Unbeknownst to participants, one deck contained a uniform distribution of numbers between 1 and 9 ('even deck'), one deck contained more 1's than other numbers ('low deck'), making gambles 30% less likely to succeed, and one deck contained more 9's than other numbers ('high deck'), making gambles 30% more likely to succeed. In the first 15 trials, the computer drew the numbers 4, 5, and 6 three times each, and the other numbers once each. To ensure that all participants gambled in approximately 50% of trials, the numbers that the computer drew three times each were increased by one (e.g., [4, 5, 6] to [5, 6, 7]), in each subsequent set of 15 trials, if subjects took two thirds or more of the gambles against these numbers in the previous 15 trials, or decreased by one if participants took a third or less of the gambles. Participants' decks were pseudorandomly ordered while ensuring that the three decks were matched against similar computers' numbers and that no deck appeared in successive trials more than the other decks.

On both sessions, the game was identical, with the only difference being subjects played with distinct sets of three decks, indicated by different colours, where colour order and colour-associated win probability randomly varied across participants (Fig. 1b). Subjects were informed that the decks from session II were entirely unrelated to the ones from session I, and they had to learn about the novel decks anew.

To familiarize participants with the basic structure of the task, subjects, on both sessions, performed a 60-trial training block with an 'even' deck, where visual feedback indicated the number that participants drew.

**Logistic regression analysis**. We fitted a trial-by-trial logistic regression model to subjects' decisions:

$$p(c_t = 1) = \frac{1}{1 + e^{-(\beta_0 + \beta_1 x_{1_t} + \beta_2 x_{2_t} + \beta_3 x_{3_t})}}, \tag{1}$$

where a subject either accepted ($c_t = 1$) or declined ($c_t = 0$) a gamble on trial t. Here, $x_{1_t}$ is the computer number, scaled to range between $-1$ (for number 9) and 1 (for number 1). $x_{2_t}$ represents cumulative success, reflecting, for the deck played with on trial t, the sum of previous positive outcomes, computed as $+1$ multiplied by the computer's number against which it was received. $x_{3_t}$ represents cumulative failure, reflecting, for the deck played with on trial t, the sum of previous negative outcomes, computed as $-1$ multiplied by (10 - computer's number), against which it was received. The multiplications by the computer's number reflect the fact that a win against a higher computer number provides stronger evidence in favour of a deck, and in a similar vein, a loss against a lower computer number provides stronger evidence against a deck. Thus, we refer to regressors as cumulative success/failure, instead of merely cumulative wins/losses. To adequately compare effect sizes between coefficients, $x_{2_t}$ and $x_{3_t}$ were range-normalized between $-1$ and 1. Positive coefficients for the first predictor ($\beta_1$) indicate that subjects were more likely to gamble against lower computer numbers. Positive coefficients for the second predictor ($\beta_2$) indicate that subjects were more likely to gamble given a deck with which they had experienced more cumulative success. Positive coefficients for the third predictor ($\beta_3$) indicate that subjects were more likely to decline a gamble given a deck with which they had experienced more cumulative failure. $\beta_0$ is the intercept.

Note that the regression did not converge for one subject on session I, thus we discarded this data from the group analysis.

**Computational modelling**

*Model space*. To account for the precise mechanisms that guided learning from reward and punishment, we compared a set of computational learning models in terms of how well each model explained subjects' choices. Note that although our task involves gambling, it contrasts with typical risky decision-making paradigms (e.g.,[54,73]) in that decision variables need to be learned by trial and error. Moreover, the typical approach in risky decision-making studies for estimating a utility function is not suitable here, since gains and losses have only one possible size. Thus, we modelled the data using a variety of reinforcement learning models, which have been shown previously to adequately capture risk sensitivity in the context of trial-and-error learning[7,10]. In all models, the probability of taking a gamble was modelled by applying a logistic function to a term that represented available evidence.

Model 1 ('gambling bias') and model 2 ('gambling bias & computer number') are oblivious to previous experience with the decks, and do not assume any learning to occur.

Here, model 1 computes the evidence as:

$$\beta' \tag{2}$$

where $\beta'$ is a gambling bias parameter, determining a subject's general propensity to gamble, thus allowing the model to favour either gambling or declining to begin with.

Model 2 computes the evidence as:

$$\beta' + \beta'' N_t \tag{3}$$

where N is the computer drawn number at trial $t$, scaled to range between $-1$ (for

number 9) and 1 (for number 1), equivalent to the logistic regression, and $\beta''$ is an inverse temperature parameter, determining the strength, with which the computer's number is determining a decision to gamble.

Model 3 ('Q-learning') learns the expected outcome of gambles with each deck $d$ as follows:

$$Q_{t+1}^{d_t} = Q_t^{d_t} + \eta \delta_t, \tag{4}$$

where

$$\delta_t = r_t - Q_t^{d_t} \tag{5}$$

is an outcome prediction error, reflecting the difference between the actual ($r_t$) and the expected ($Q_t^{d_t}$) outcome of a gamble (initialized as $Q_0^{d_0} = 0$). $r_t = 1$ represents a win, and $r_t = -1$ represents a loss, and $\eta$ is a learning rate parameter that weights the influence of prediction errors on subsequent expectations. Model 3 then computes the evidence as:

$$\beta' + \beta'' N_t + \beta''' Q_t^{d_t}, \tag{6}$$

where $\beta'''$ is a free parameter that determines the strength, with which choices are directed towards higher Q-value options.

In contrast to the previous model, model 4 ('adjusted Q-learning') computes prediction errors with respect to expectations that additionally factor in the computer's number:

$$\delta_t = r_t - Q_t^{d_t} - N_t, \tag{7}$$

which means the model learns more from more surprising outcomes, i.e., from win outcomes of gambles against higher numbers, and from loss outcomes of gambles against lower numbers.

Based on prior work[7,11], we assumed subjects would learn at a different rate from successful, i.e., reward, and unsuccessful gambles, i.e., punishment. In contrast to the gambling bias parameter ($\beta'$) that was included in all models, allowing them to favour either gambling or declining to begin with, an asymmetric learning bias can make such a tendency evolve with learning over time.

To this end, model 5 ('asymmetric Q-learning') and model 6 ('adjusted & asymmetric Q-learning') incorporate two distinct learning rate parameters ($\eta^+$ & $\eta^-$), that allow learning at a different rate from different outcome types, i.e., from wins:

$$Q_{t+1}^{d_t} = Q_t^{d_t} + \eta^+ \delta_t, \tag{8}$$

and from losses:

$$Q_{t+1}^{d_t} = Q_t^{d_t} + \eta^- \delta_t \tag{9}$$

Note that a model with different positive and negative learning rates for each deck could not be estimated due to the number of outcomes subjects observed varying substantially across decks and outcome type, such that not all subjects observed both positive and negative outcomes for each of the decks. Thus, in accordance with our earlier work using an equivalent task[7], we assumed the same two learning rates characterized learning about all decks. We acknowledge a limitation of this approach is that learning rate estimation is more heavily influenced by trials from the high, followed by the even and then the low deck, as subjects gambled more often with better decks and consequently observed more outcomes from which they could learn.

*Model fitting*. To fit the parameters of the different models to subjects' decisions, we used an iterative hierarchical expectation-maximization procedure across the entire sample, separately for each session[56,74]. We sampled $10^5$ random settings of the parameters from predefined prior distributions. Then, we computed the likelihood of observing subjects' choices given each setting and used the computed likelihoods as importance weights to re-fit the parameters of the prior distributions. These steps were repeated iteratively until model evidence ceased to increase. To derive the best-fitting parameters for each individual subject, we computed a weighted mean of the final batch of parameter settings, in which each setting was weighted by the likelihood it assigned to the individual subject's decisions. Note that the hierarchical fitting procedure, including all priors, were applied to the entire sample without distinguishing between SSRI and placebo subjects. This ensured that the parameter estimates, at the level of individual subjects, were mutually independent given the shared prior, rendering it appropriate to assess between-group differences. Learning rate parameters ($\eta$, $\eta^+$ & $\eta^-$) were modelled with Beta distributions (initialized with shape parameters $a = 1$ and $b = 1$). The gambling bias parameter ($\beta'$) was modelled with a normal distribution (initialized with $\mu = 0$ and $\sigma = 1$), and inverse temperature parameters ($\beta''$ & $\beta'''$) were modelled with Gamma distributions (initialized with $\kappa = 1$, $\theta = 1$).

*Model comparison*. We compared between models in terms of how well each model accounted for subjects' choices by means of the integrated Bayesian Information Criterion (iBIC[56,75]). Here, we estimated the evidence in favour of each model ($\lambda$) as the mean likelihood of the model given $10^5$ random parameter settings drawn from the fitted group-level priors. We then computed the iBIC by penalizing the model evidence to account for model complexity as follows: iBIC $= -2 \ln \lambda + \kappa \ln$ n, where $\kappa$ is the number of fitted parameters, and n is the total number of subject

choices used to compute the likelihood. Lower iBIC values indicate a more parsimonious model fit.

**Statistics and reproducibility.** Drug effects were assessed using repeated measures analyses of variance (rm-ANOVA) and independent samples t-tests. Our sample size ($n = 66$) was similar to related studies related using a comparable pharmacological protocol, e.g.,[26,28]. We did not attempt to reproduce the pharmacological results. However, the results of our study on learning asymmetries replicate previous findings using an identical cognitive task[7].

**Reporting summary.** Further information on research design is available in the Nature Research Reporting Summary linked to this article.

## Data availability

The data analysed during this study are available on GitHub (https://github.com/jmichely/ssri_asymmetric_learning) and Zenodo (https://zenodo.org/badge/latestdoi/262273939). Supplementary Data 1 contains all source data underlying the graphs and charts presented in the main figures of the manuscript.

## Code availability

The custom computer code used to generate the reported results are available GitHub (https://github.com/jmichely/ssri_asymmetric_learning) and Zenodo (https://zenodo.org/badge/latestdoi/262273939).

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

## Acknowledgements
J.M. was supported by a fellowship from the German Research Foundation (MI 2158/1-1) and is participant in the BIH Charité (Junior) (Digital) Clinician Scientist Program funded by the Charité – Universitätsmedizin Berlin, and the Berlin Institute of Health at Charité (BIH). R.J.D. holds a Wellcome Trust Investigator award (098362/Z/12/Z). The Max Planck UCL Centre for Computational Psychiatry and Ageing Research is a joint initiative supported by the Max Planck Society and University College London. The Wellcome Centre for Human Neuroimaging is supported by core funding from the Wellcome Trust (091593/Z/10/Z).

## Author contributions
J.M. and E.E. designed the experiments. J.M. and I.M.M. performed the experiments. J.M., E.E., A.E. and R.J.D. analysed and interpreted the data. J.M., E.E. and R.J.D. wrote the paper.

## Funding

## Competing interests
The authors declare no competing interests.
