## [Peer Review File · Communications Biology]

Reviewers' comments:

Reviewer #1 (Remarks to the Author):

Review for "serotonin modulates asymmetric learning from reward and punishment" by Michely and co. The authors investigate a very debated topic concerning the pharmacological bases of reinforcement learning: what is the role of serotonin in reward vs. punishment learning? In the paper they present the results of a placebo controlled experiment using SSRI, where subjects are tested twice: at baseline and after one week of treatment. The groups did not differ at baseline, but they do after one week of treatment in a way that is consistent with the idea that 5HT boosts punishment learning. This is a well conducted study that will contribute to the ongoing debate concerning the role of 5HT in RL, which has both a fundamental and applied interest. I have few suggestions to make.

1. Even if I do not think that this critically undermines the interest of the paper, it should be explicitly acknowledged that the absence of triple interaction (drug, session, valence) represents a limitation and, accordingly, some of the claims should be toned down a bit. The factor "session" should also be entered in the statistical model of the regression coefficients.

2. In session 1 there is a clear effect of valence in the learning rates (where positive valence is higher compared to negative in both groups). This finding replicates Lefebvre et al. NHB 2017. I think the authors should mention this and discuss their results in the light of the fact that inter-individual differences in the learning rate asymmetry has been associated with striatal hyperactivity in the above mentioned study. How does it fit with what is known about 5HT function and distribution in the striatum?

3. Supplementary Fig. S1 delivers an important message. It should be put in the main text.

Reviewer #2 (Remarks to the Author):

Review of Serotonin modulates asymmetric learning from reward and punishment by Michely et al

Overview:

The authors investigate how a week-long administration of SSRI citalopram influence learning in the context of a betting task which involves both rewards and punishments. The topic is very interesting and timely. Nevertheless, the paper needs more clarification on modelling choices, presentation of data and interpretation. Below, I'm providing a detailed list to the authors and hope that they will take these recommendations constructively.

Recommendations to the authors

1-Considering the modelling emphasis in this paper, I would appreciate if the authors submit a revised version of the paper with line numbers, numbered equations and formal mathematical descriptions of all the logit/logistic regression models. These would make it easier to give more precise recommendations.

2-Figure 1C and D it is not clear whether these percentages come from the citalopram or the placebo groups or if they are the population averages?

3-one of the key problems is the interpretation of findings shown in figure 2. A common problem across all the figures is that the authors did not communicate whether the error bars reflect SD or SEM? More specifically for Figure 2, the panel legends did not appear on the figure (ABCD), but I assumed these should be from left to right.

Somehow, the authors did not conduct the repeated measures ANOVA which also includes the data from the 1st visit, as they have done for the model-based analysis. This is an omission which stands out. Looking at the data in detail, I do not agree with the authors interpretation that SSRI treatment enhanced gambling avoidance as a function of cumulative failure. The true test of this would be to compare SSRI group from day one to day 7, which the authors did not. Ideally, one would like to see the behaviour from the placebo group remaining stable and drug group changing significantly in one direction. Currently, the differences observed between the groups on day 7 is mostly driven by the placebo group acting differently which augment the differences between the

groups. I am well aware that it is rather difficult to conduct these within subject experiments, but the interpretation could have been much clearer if the authors had another session one week earlier than the current visit 1. At least then, one could evaluate whether weight reductions observed in the placebo group is a function of global learning about the task environment, such that the weights decrease consistently across different assessment points. The current interpretation is stronger than what is shown in the data.

4-I am little bit confused about the way cumulative success failure are computed. In the methods, the authors say that, this is computed by +1 multiplied by the computers number against which it was received. I do not follow the logic of this multiplication, and the authors need to provide a clear example including the justification why this has been done in a multiplicative manner.

5-another key problem could be that the way these regressors are coupled. For example, if a participant is consistently accepting to gamble, either winning or losing money resulting from these gambles, cumulative failures would be equivalent to total number of trials minus total number of win outcomes (as one can only win or lose). In order to rule out any correlation between the regressors, I think the authors need to provide covariance matrices for all the regressors.

6-although, cumulative success or failure could be one index of learning happening during the task, using the outcomes of the t-1st trial as regressors could be a more intuitive way of unifying model free and model-based analyses, as learning rates captured by the model-based analysis based on behavioural update from the prediction errors arising from one previous trial.

7- as far as I understood, learning rates were estimated from all of the blocks all at once.

Although, this approach may look okay, it brings about an inherent problem about understanding what serotonin actually does. This is because, as shown in figure 1C and D, participants choose to gamble more in the high deck relative to the lower deck. For the even deck, if the reward probabilities are 50%, then actually there's nothing to learn from this environment (i.e. maximum entropy, pure noise), so learning rates associated with these trials should diminish. Unless outcomes of the declined gamble choices are also used for generating prediction errors (this information should be included explicitly, and again this condition is also 50%), the current learning rates are estimated in a biased fashion relying more to the trials from the higher deck, followed by the even deck and little bit from the low deck as participants usually opt out from this condition. This approach is rather unspecific. I am wondering whether it would be possible to estimate learning rates for each type of deck, similar to modelling Pavlovian Instrumental Learning Task as in that task the conditions are also interleaved. Either way, the authors need to provide a much-detailed description of such confounding factors that I highlighted above. Currently, the paper is lacking important details surrounding these issues.

8-considering the current task involves participants making betting decisions, I am wondering why the authors did not consider risk and loss aversion in their models? I think there needs to be a clear justification as to why the task should be modelled in terms of reinforcement learning rather than risky decision-making, or ideally both. The introduction probably should mention the literature involving serotonergic agents acting on risk seeking tendencies.

Addressing these issues may substantially change the authors results and interpretations in the discussion section. I would be happy to review a revised version of this manuscript in due course. I hope the authors will take these recommendations constructively.

Reviewer #3 (Remarks to the Author):

Michely and colleagues performed a double-blind placebo-controlled study in sixty-six healthy participants to investigate whether modulating central 5HT-levels affects how much learning is driven by reward and punishment. Half of the participants were administered the SSRI citalopram for seven consecutive days, the other half were given placebo, and all participants performed a gambling task on day 1 and 7. Compared to placebo, citalopram enhanced learning from negative feedback (monetary losses) and diminished learning from positive feedback (monetary gains) on day 7.

This is a nice piece of work which was well designed and thoroughly conducted and analysed. It provides a simple and clear message that is communicated well. I have a few issues with the

analyses and interpretation that should be addressed.

Major:

(1) The authors are framing their results in a way that might explain why SSRI treatment effects evolve slowly. When framed in terms of the effect asymmetric learning rates have on expectations, the direction of their effects (enhanced learning from negative, diminished learning from positive outcomes) seems intuitive: when expectations are increasingly negative, experiences, by comparison, are overall more positive than expected. However, it becomes more difficult to try and reconcile the results of the present study with the wider literature. For example, there is a body of work showing negative biases are reduced following SSRI treatment. Also, it is well established that healthy people have an optimism bias, they learn more from positive information relative to negative information. However, following the author's logic, healthy people should become depressed because with higher expectations, experiences should be generally disappointing. Furthermore, the optimism bias is not present in depressed patients (Garrett, ... Dolan, 2015; Korn ... Dolan 2013), where learning from negative and positive feedback is more balanced. The results put forward here suggest that SSRIs frequently used to treat depression do not normalize these learning asymmetries in the direction that would move patients towards the healthy population (increasing learning from positive outcomes), but instead push them further away from the healthy population. In other words, negative learning biases thought to contribute to depression are pronounced further after seven days on SSRIs, but the authors argue this could help patients' mood in the long run. This seems counterintuitive and needs further explanation.

(2) I have a few comments related to the logistic regression of the choice data (Figure 2) and its relationship to the modelling results (Figure 3).

a. Despite high correlations between learning asymmetries captured with the logistic regression and the two learning rates from the RL model (Figure 3 - supplement 2), it seems that these two analyses are not necessarily measuring the same behaviour. In an RL model, the strongest weight is given to the most recent outcomes but the cumulative losses regressor does not contain any recency weighting. To make the comparison clearer, the logistic regression could be estimated using recency-weighted cumulative gains/losses, or even simply include the last outcome for the current deck as an additional regressor. It seems that the reason why the comparison looks strong here (Fig 3 - S2, high correlations) is because learning rates are extremely low (Figure 3), so the recency-weighting is relatively flat. Nevertheless, it cannot be assumed that in general, the logistic regression's cumulative regressor and the RL model's learning rate measure the same behavioural influence.

b. Relatedly, "cumulative wins" is a somewhat misleading term given it is not merely measuring the cumulative wins (or losses). Wins (losses) are multiplied by the number shown on the computer's deck, so the resulting regressors rather capture how lucky or unlucky the participant has been. (Note, the final RL model also contains this term in the prediction error.)

c. The logistic regression should include interaction effects. For example, one might expect an interaction between computer's drawn number and deck type. It would also be interesting to examine whether losses in general versus losses received for the present card deck (i.e. respecting correct credit assignment, as currently coded, or not) are driving the asymmetries seen following citalopram administration. This could be distinguished by comparing two models, one containing cumulative wins/losses for the current card deck, the other containing cumulative wins/losses across all card decks.

(3) The full interaction session x drug x valence is not significant. To conclude that the drug effect was specific to day 7, however, it would be necessary to show an interaction with session which is currently missing. I wondered whether the session x drug interaction is significant for either positive or negative learning rates (or both) in Figures 2 + 3?

(4) Have the authors tried a model with a positive and negative learning rate per card deck (a total of six learning rates)? I don't know if such a model would be estimable given the number of trials, but this can be checked using simulations. This is interesting because the distribution of the numbers drawn by different decks differs, and this might affect their respective uncertainty/volatility estimate, which we know from Pulcu & Browning's work affects the learning rate.

(5) The parameter recovery is either wrong or its description is inaccurate. The standard is to produce choices from an artificial agent with a chosen set of parameters and then plot those chosen parameters ('ground truth') against the parameters fitted to the choices produced by this artificial agent ('recovered parameters'). This is then repeated by systematically varying all parameters in the expected range. However, Suppl Fig 4 seems to show a relationship between simulated and real data, rather than between ground truth and recovered (both relating to the synthetic data). This is either a reporting mistake or a more fundamental error, but establishing reliable parameter recovery is critical.

Minor Comments:

(1) What was the rationale for completing the first behavioural session after a single dose of citalopram rather than before any citalopram administration?

(2) Is it the case that the first two regressors are scaled between -1 and 1 in Figure 2 (logistic regression) and the next two (cumulative wins/losses) between 1-9? This won't affect any conclusions, but it is standard to range-normalize or z-score regressors to make effect sizes comparable.

(3) The dopamine-serotonin opponency theory is surprisingly prominent given it is no longer considered timely.

(4) Please report the optimal learning rate in this task. Intuitively, it seems that slower learning is better than faster learning given the card decks have a constant number distribution. This could change the interpretation of learning rate changes seen in Figure 3.

(5) Did the authors test a model with outcome sensitivity parameters (for gains and losses)? Could sensitivity to outcomes rather than changes in updating be explaining the results?

(6) Please add individual data points to figures.

(7) Please report precise p-values, throughout.

(8) Figure 2 is missing labels A-D.

We thank the reviewers for the positive evaluation of our manuscript “Serotonin modulates asymmetric learning from reward and punishment” (COMMSBIO-20-3363-T). We are grateful for the opportunity to revise the manuscript and believe the revised version is substantially improved in light of the reviewers’ comments.

We addressed all comments point-by-point. The respective changes in the manuscript are marked in bold. Please note that the page numbers given below refer to the page numbers of the revised manuscript.

Reviewer #1 (Remarks to the Author):

Review for “serotonin modulates asymmetric learning from reward and punishment” by Michely and co. The authors investigate a very debated topic concerning the pharmacological bases of reinforcement learning: what is the role of serotonin in reward vs. punishment learning? In the paper they present the results of a placebo controlled experiment using SSRI, where subjects are tested twice: at baseline and after one week of treatment. The groups did not differ at baseline, but they do after one week of treatment in a way that is consistent with the idea that 5HT boosts punishment learning. This is a well conducted study that will contribute to the ongoing debate concerning the role of 5HT in RL, which has both a fundamental and applied interest. I have few suggestions to make.

We thank the reviewer for a positive evaluation of our study and the constructive suggestions that we address below.

1) Even if I do not think that this critically undermines the interest of the paper, it should be explicitly acknowledged that the absence of triple interaction (drug, session, valence) represents a limitation and, accordingly, some of the claims should be toned down a bit. The factor “session” should also be entered in the statistical model of the regression coefficients.

We thank the reviewer for the suggestion to test for a three-way interaction (drug x session x valence) for the regression results. We have now implemented this in both the model-agnostic analysis of gambling preference, and in the more sensitive model-based analysis of learning rates.

In the former, the pharmacological effect on gambling preferences as a function of outcome type did not reach statistical significance (drug x valence, $p=0.108$; drug x session x valence, $p=0.124$). We have added this analysis to the revised manuscript.

“However, when we assessed the data across both sessions, the pharmacological effect on gambling preferences as a function of outcome type did not reach statistical significance (drug: $F_{1,63}=1.7$, $p=0.194$, drug x valence: $F_{1,63}=2.6$, $p=0.108$, drug x session x valence: $F_{1,63}=2.4$, $p=0.124$).” (page 6/7, results)

In the model-based analysis of learning rates, the drug x valence interaction was significant, but the three-way interaction was not:

“We found a significant asymmetric effect on learning rates across data from both sessions (drug x valence: $F_{1,62}=4.1$, $p=0.046$; drug $F_{1,62}=0.8$, $p=0.365$), but no significant

three- way interaction (drug x valence x session: $F_{1,62}=1.0$, $p=0.305$, controlling for an overall gambling bias, Supplementary Fig. S1).” (page 8, results)

We agree with the reviewer that it is critical to acknowledge the absence of a three-way interaction. We duly downtone the interpretation and mention this limitation explicitly in the revised manuscript, where we also outline how more complex study designs might address this limitation in the future:

“Although our data suggests an emergence of serotonergic effects after a temporally extended intervention, a three-way interaction (drug x session x valence) was not significant. Thus, we can conclude the drug had an hypothesized effect, but not an effect that was significantly greater after prolonged (week-long, on day 7) as compared to acute (single-dose, day 1) treatment. To unravel the precise trajectory of any such effect, future studies will ideally include a pre-drug testing session as well as multiple sessions over several weeks of treatment.” (page 13, discussion)

2) In session 1 there is a clear effect of valence in the learning rates (where positive valence is higher compared to negative in both groups). This finding replicates Lefebvre et al. NHB 2017. I think the authors should mention this and discuss their results in the light of the fact that inter-individual differences in the learning rate asymmetry has been associated with striatal hyperactivity in the above mentioned study. How does it fit with what is known about 5HT function and distribution in the striatum?

The reviewer is right that, in session I, across all subjects, we find greater learning rates for positive, as compared to negative, outcomes. However, this effect is, across all subjects, reversed in session II; a reversal that is strongly driven by the impact of an SSRI intervention. Overall, across all subjects and sessions, there is no significant difference between positive and negative learning rates ($p=0.151$). However, in placebo subjects alone we found a statistical trend for greater positive, as compared to negative, learning rates across sessions ($p=0.063$), in line with the results by Lefebvre et al., 2017, *Nat Hum Behav*. We now include this result in the revised manuscript.

“Note that, across all subjects and sessions, we found no significant difference between positive and negative learning rates ($F_{1,65}=2.1$, $p=0.151$). However, in placebo subjects alone, we found a statistical trend for greater learning from positive as compared to negative outcomes ($F_{1,32}=3.7$, $p=0.063$). This is in line with recent work showing a learning asymmetry towards greater updating from positive information healthy individuals, without pharmacological intervention (Lefebvre et al., 2017).” (page 9, results)

Unfortunately, we were not in a position to collect neural data in our current study. However, we agree that a relationship between striatal activity and variability in learning from reward and punishment has been shown by Lefebvre et al., 2017, *Nat Hum Behav*, and also in a prior study of ours (Eldar et al., 2016, *PNAS*). In light of the reviewer’s comment, we have revised our manuscript to address putative neurobiological underpinnings of the effects we highlight. Note that Macoveanu, 2014, *Brain Res*, indicates a meta-analysis of serotonergic effects on the neural basis of human reward and punishment processing.

“Moreover, learning asymmetries are linked to interindividual variability in brain structure and function (Eldar et al., 2016; Lefebvre et al., 2017), and are thought to play a

role in the emergence of mood disorders, often characterised in terms of aberrant processing of reward and punishment (Murphy et al., 2003; Eshel & Roiser, 2010).” (page 2, introduction)

“Unfortunately, in this study, we were not in a position to acquire neural data. In light of previous studies on interindividual variability in human learning asymmetries, it is tempting to speculate that serotonergic agents may act preferentially in the striatum and prefrontal cortex to alter the relative degree of impact from positive and negative outcomes (Macoveanu, 2014; Eldar et al., 2016; Lefebvre et al., 2017).” (page 9, discussion)

3) Supplementary Fig. S1 delivers an important message. It should be put in the main text.

We thank the reviewer for this constructive suggestion. We have moved the respective figure to the main manuscript (cf. Fig. 4 in the revised manuscript), as we agree it improves understanding of the manuscript’s key message.

Reviewer #2 (Remarks to the Author):

Review of Serotonin modulates asymmetric learning from reward and punishment by Michely et al.

Overview:

The authors investigate how a week-long administration of SSRI citalopram influence learning in the context of a betting task which involves both rewards and punishments. The topic is very interesting and timely. Nevertheless, the paper needs more clarification on modelling choices, presentation of data and interpretation. Below, I’m providing a detailed list to the authors and hope that they will take these recommendations constructively.

We thank the reviewer for the constructive comments that enabled us to improve our manuscript. Below we address the reviewer’s proposed suggestions.

Recommendations to the authors

1) Considering the modelling emphasis in this paper, I would appreciate if the authors submit a revised version of the paper with line numbers, numbered equations and formal mathematical descriptions of all the logit/logistic regression models. These would make it easier to give more precise recommendations.

We apologise for an insufficient description of our regression model. In the revised manuscript, we now provide a more detailed presentation, including mathematical equations for our regression model (in line with the presentation of our computational models).

$$p(c_t = 1) = \frac{1}{1 + e^{-(\beta_0 + \beta_1 x_{1t} + \beta_2 x_{2t} + \beta_3 x_{3t})}}$$

, where a subject either accepted ($c_t=1$) or declined ($c_t=0$) a gamble on trial t . Here, x_{1t} is the computer number, scaled to range between -1 (for number 9) and 1 (for number 1). x_{2t} represents cumulative success, reflecting, for the deck played with on trial t , the sum of previous positive outcomes, computed as +1 multiplied by the computer's number against which it was received. x_{3t} represents cumulative failure, reflecting, for the deck played with on trial t , the sum of previous negative outcomes, computed as -1 multiplied by (10 - computer's number), against which it was received. The multiplications by the computer's number reflect the fact that a win against a higher computer number provides stronger evidence in favour of a deck, and in a similar vein, a loss against a lower computer number provides stronger evidence against a deck. Thus, we refer to regressors as cumulative "success/failure", instead of merely cumulative "wins/losses". To adequately compare effect sizes between coefficients, x_{2t} and x_{3t} were range-normalized between -1 and 1. Positive coefficients for the first predictor (β_1) indicate that subjects were more likely to gamble against lower computer numbers. Positive coefficients for the second predictor (β_2) indicate that subjects were more likely to gamble given a deck with which they had experienced more cumulative success. Positive coefficients for the third predictor (β_3) indicate that subjects were more likely to decline a gamble given a deck with which they had experienced more cumulative failure. β_0 is the intercept." (page 17, methods)

2) Figure 1C and D it is not clear whether these percentages come from the citalopram or the placebo groups or if they are the population averages?

In the initial version of our manuscript, these percentages represented population averages. However, in light of the reviewer's comment, we have now revised the figure (in line with the other figures, where results are presented separately for the two groups), which now shows results for SSRI and placebo groups, respectively (cf. novel Fig. 1C/D below).

3) one of the key problems is the interpretation of findings shown in figure 2. A common problem across all the figures is that the authors did not communicate whether the

error bars reflect SD or SEM? More specifically for Figure 2, the panel legends did not appear on the figure (ABCD), but I assumed these should be from left to right.

We apologise for lack of clarity on this. Error bars reflect SEM throughout, and this is now mentioned in all figure legends. Additionally, we have added panel labels in Fig. 2.

Somehow, the authors did not conduct the repeated measures ANOVA which also includes the data from the 1st visit, as they have done for the model-based analysis. This is an omission which stands out. Looking at the data in detail, I do not agree with the authors interpretation that SSRI treatment enhanced gambling avoidance as a function of cumulative failure. The true test of this would be to compare SSRI group from day one to day 7, which the authors did not. Ideally, one would like to see the behaviour from the placebo group remaining stable and drug group changing significantly in one direction. Currently, the differences observed between the groups on day 7 is mostly driven by the placebo group acting differently which augment the differences between the groups. I am well aware that it is rather difficult to conduct these within subject experiments, but the interpretation could have been much clearer if the authors had another session one week earlier than the current visit 1. At least then, one could evaluate whether weight reductions observed in the placebo group is a function of global learning about the task environment, such that the weights decrease consistently across different assessment points. The current interpretation is stronger than what is shown in the data.

We thank the reviewer for the suggestion to compute a three-way interaction (session x group x valence) for the regression results. The results trended towards a drug x valence interaction ($p=0.108$), and a three-way interaction ($p=0.124$), but were not significant. We have added this analysis to the revised manuscript.

“However, when we assessed the data across both sessions, the pharmacological effect on gambling preferences as a function of outcome type did not reach statistical significance (drug: $F_{1,63}=1.7$, $p=0.194$, drug x valence: $F_{1,63}=2.6$, $p=0.108$, drug x session x valence: $F_{1,63}=2.4$, $p=0.124$).” (page 6, results)

As mentioned for the model-based analysis of learning rates, however, the drug x valence interaction was significant, but the three-way interaction was not:

“We found a significant asymmetric effect on learning rates across data from both sessions (drug x valence: $F_{1,62}=4.1$, $p=0.046$; drug $F_{1,62}=0.8$, $p=0.365$), but no significant three-way interaction (drug x valence x session: $F_{1,62}=1.0$, $p=0.305$, controlling for an overall gambling bias, Supplementary Fig. S2).” (page 7, results)

Here, however, it is critical to keep in mind that both the day 1 and the day 7 sessions were conducted under the influence of the drug. Therefore, the significant two-way interaction that we report for the model-based results shows an overall drug effect. What cannot be concluded is that this effect is greater on day 7 (week-long) than on day 1 (single-dose).

Moreover, the data are inconsistent with a suggestion that the results reflect changes in the placebo group. Overall, learning rates, irrespective of valence, do not change significantly between sessions in the placebo group ($p=0.060$). Additionally, a session x valence interaction is clearly significant in SSRI subjects ($p<0.001$), and not in placebo

subjects ($p=0.054$). Thus, the evidence points to a difference between groups being driven by effects in the SSRI group.

We agree with the reviewer that adding a baseline (pre-drug) session would have been helpful, but unfortunately, we were not in a position to do so for practical reasons. Importantly, however, due to the nature of a placebo-controlled between-subject drug design, any differences between study groups (including those observed on day 7) can only be attributed to the drug intervention. Thus, overall, we show evidence for an asymmetric effect of drug on positive and negative learning rates. We, take the concerns raised by the reviewer seriously and now add the additional analysis as well as a detailed limitation discussion section to the revised manuscript, where we address this particular issue and propose how future studies can better address it:

“Although our data suggests an emergence of serotonergic effects after a temporally extended intervention, a three-way interaction (drug \times session \times valence) was not significant. Thus, we can conclude the drug had an hypothesized effect, but not an effect that was significantly greater after prolonged (week-long, on day 7) as compared to acute (single-dose, day 1) treatment. To unravel the precise trajectory of any such effect, future studies will ideally include a pre-drug testing session as well as multiple sessions over several weeks of treatment.” (page 13, discussion)

4) I am little bit confused about the way cumulative success failure are computed. In the methods, the authors say that, this is computed by +1 multiplied by the computers number against which it was received. I do not follow the logic of this multiplication, and the authors need to provide a clear example including the justification why this has been done in a multiplicative manner.

The regressors quantify the evidence provided by different wins and loss, respectively. A win against a high computer number, for instance 7, provides more evidence that the deck with which one played contains high numbers. Specifically, in this case it would mean that the card drawn from the deck was either an 8 or 9. By contrast, winning against a low number, for instance 2, would only mean that we drew a number that is 3 or greater. Thus, the latter win does not provide as much evidence in favour of our deck. Similarly, losing against a low number provides more evidence against our deck compared to losing against a high number. This computation mirrors the computation of prediction errors in the winning computational reinforcement learning model.

We now clarify this in the revised version of the manuscript, explaining the rationale for referring to this term as cumulative “success/failure”, instead of merely cumulative “wins/losses”.

“ x_{2t} represents cumulative success, reflecting, for the deck played with on trial t , the sum of previous positive outcomes, computed as +1 multiplied by the computer's number against which it was received. x_{3t} represents cumulative failure, reflecting, for the deck played with on trial t , the sum of previous negative outcomes, computed as -1 multiplied by (10 - computer's number), against which it was received. The multiplications by the computer's number reflect the fact that a win against a higher computer number provides stronger evidence in favour of a deck, and similarly, a loss against a lower computer number provides stronger evidence against a deck. Thus, we refer to the regressors as cumulative “success/failure”, instead of merely cumulative “wins/losses”. To adequately compare effect

sizes between coefficients, x_{2_t} and x_{3_t} were range-normalized between -1 and 1.” (page 17, methods)

5) another key problem could be that the way these regressors are coupled. For example, if a participant is consistently accepting to gamble, either winning or losing money resulting from these gambles, cumulative failures would be equivalent to total number of trials minus total number of win outcomes (as one can only win or lose). In order to rule out any correlation between the regressors, I think the authors need to provide covariance matrices for all the regressors.

The reviewer raises an important issue here. We computed the covariance matrices for the regressors, and this highlighted a redundant regressor that we mistakenly included in the analysis. Specifically, ‘deck type’ (high/even/low) is in fact highly correlated with ‘cumulative success/failure’, which is to be expected as subjects experience more success with higher decks, and more failures with lower decks.

Thus, we have now revised the regression, eliminating the regressor ‘deck type’ altogether, as its effect is captured by the ‘cumulative success/failure’ regressors. Reassuringly, the revised regression shows equivalent results. Specifically, the main result of the regression analysis, an interaction (drug x valence) on session II is highly significant ($p=0.0018$), driven by a greater impact of failure ($p=0.024$), but not success ($p=0.892$), on choices in SSRI, as compared to placebo, subjects. Note that the regression did not converge for one subject on session I, thus we discarded these data from the group analysis. Critically, including or removing this one data set did not impact any of our findings.

We thank the reviewer for this suggestion, which helped us correct an error and improve the clarity of our methods and results. We have revised the manuscript accordingly.

“Next, we used a trial-by-trial logistic regression approach (cf. Materials and Methods) to assess whether subjects’ decisions to gamble were dependent upon previous receipt of positive, or negative, outcomes over time. First, we found that subjects, over both sessions, gambled more against lower computer numbers (session I: $t_{64}=14.7$, $p<0.001$; session II: $t_{65}=20.9$, $p<0.001$). Second, participants, over the course of each session, gambled more with decks, with which they had experienced more success (session I: $t_{64}=10.3$, $p<0.001$; session II: $t_{65}=15.7$, $p<0.001$), and less with decks, with which they had experienced more failure (session I: $t_{64}=10.2$, $p<0.001$; session II: $t_{65}=11.0$, $p<0.001$). This result indicates subjects successfully learned about the decks from the outcomes of their gambles.

On session I, effects were similar across drug groups for cumulative success and failure (drug x valence: $F_{1,63}=0.3$, $p=0.844$; drug: $F_{1,63}=0.9$, $p=0.345$; Fig. 2A). At session II, however, we found evidence for an asymmetric impact of outcome success and failure, as a function of treatment (drug x valence, $F_{1,64}=10.5$, $p=0.0018$; drug: $F_{1,64}=2.4$, $p=0.126$; Fig. 2B), attributable to an enhanced impact of failure ($t_{64}=2.3$, $p=0.024$) but not of success ($t_{64}=0.1$, $p=0.892$), in SSRI treated as compared to placebo subjects. This differential pattern suggests that SSRI treatment increased an impact of negative outcomes, enhancing a gamble avoidance tendency in response to failure.” (page 5-7, results)

“Note that the regression did not converge for one subject on session I, thus we discarded this data from the group analysis.” (page 17, methods)

Figure 2. Results of trial-by-trial logistic regression model.

Fitting a logistic regression model to subjects' decisions showed that participants gambled more against lower computer numbers, with no drug differences on session I (A), and session II (B), respectively. Additionally, subjects, over the course of a session, gambled more with increasing success with each deck, and gambled less with increasing failure with each deck. On session I, impact of cumulative success and failure was unaffected by treatment (A). On session II, however, SSRIs enhanced the impact of failure but not wins (B), indicating an asymmetric drug effect on reward and punishment. ** $p < 0.01$, * $p < 0.05$, n.s.=not significant. Error bars indicate SEM.

6) although, cumulative success or failure could be one index of learning happening during the task, using the outcomes of the t-1st trial as regressors could be a more intuitive way of unifying model free and model-based analyses, as learning rates captured by the model-based analysis based on behavioural update from the prediction errors arising from one previous trial.

The reviewer raises an interesting issue here, and we have attempted to implement this suggestion. However, in doing so we found that our experimental design is not suited to an analysis that relies on behavioural change across consecutive trials, i.e., from trial $t-1$ to trial t for the following reason.

In our experiment, if a subject declines a gamble, e.g., on trial $t-1$, then there is nothing to be learnt from, as outcomes from declined gambles are not shown to participants. Thus, a switch from a declined gamble to a gamble, cannot be predicted based on trial $t-1$. In contrast, behavioural change is more likely to arise from a change in computer number. Critically, learning mostly manifests in people continuing to gamble despite a higher computer number, but due to variability in the computer numbers this would not manifest cleanly in an analysis that focuses on outcomes on trial $t-1$. Thus, given our specific experimental design, such an analysis has limited value.

We have, however, followed a related suggestion by reviewer #3, where we conducted an additional analysis computing a regression that includes the last outcome for the current deck as an additional regressor, in addition to the running outcome regressor over the course of the experiment. This regression model, however, fits worse (cf. response #3.2), than our original regression, indicating that recency did not play a major role in explaining subjects' choices.

We realised, however, based on the reviewer's suggestion, that it is important to clarify precisely how the regression diverges from the computational model, and what is the purpose of each analysis. We have now revised the manuscript accordingly:

“Next, we used a trial-by-trial logistic regression approach (cf. Materials and Methods) to assess whether subjects’ decisions to gamble were dependent upon previous receipt of positive, or negative, outcomes over time. First, we found that subjects, over both sessions, gambled more against lower computer numbers (session I: $t_{64}=14.7$, $p<0.001$; session II: $t_{65}=20.9$, $p<0.001$). Second, participants, over the course of each session, gambled more with decks with which they had experienced more success (session I: $t_{64}=10.3$, $p<0.001$; session II: $t_{65}=15.7$, $p<0.001$), and less with decks with which they had experienced more failure (session I: $t_{64}=10.2$, $p<0.001$; session II: $t_{65}=11.0$, $p<0.001$). This result indicates subjects successfully learned about the decks from the outcomes of their gambles.

On session I, effects were similar across drug groups for cumulative success and failure (drug \times valence: $F_{1,63}=0.3$, $p=0.844$; drug: $F_{1,63}=0.9$, $p=0.345$; Fig. 2A). At session II, however, we found evidence for an asymmetric impact of outcome success and failure, as a function of treatment (drug \times valence, $F_{1,64}=10.5$, $p=0.0018$; drug: $F_{1,64}=2.4$, $p=0.126$; Fig. 2B), attributable to an enhanced impact of failure ($t_{64}=2.3$, $p=0.024$) but not of success ($t_{64}=0.1$, $p=0.892$), in SSRI treated as compared to placebo subjects. This differential pattern suggests that SSRI treatment increased an impact of negative outcomes, enhancing a gamble avoidance tendency in response to failure.”

Next, we used computational modelling (cf. Materials and Methods) to assess the precise learning mechanism underlying the asymmetric effects of success and failure evident in the regression analysis. [...]“ (page 5-7, results)

“Additionally, an asymmetric effect of cumulative success and failure on gambling, as derived from the logistic regression, correlated significantly with an asymmetry in learning, as derived from the computational model (session I: $r=0.850$, $p<0.001$, session II: $r=0.841$, $p<0.001$; Fig. 4). Thus, these analyses jointly indicate that altered gambling preference, reflecting asymmetric learning from positive and negative outcomes, was influenced by serotonergic intervention.” (page 8, results)

7) as far as I understood, learning rates were estimated from all of the blocks all at once. Although, this approach may look okay, it brings about an inherent problem about understanding what serotonin actually does. This is because, as shown in figure 1C and D, participants choose to gamble more in the high deck relative to the lower deck. For the even deck, if the reward probabilities are 50%, then actually there’s nothing to learn from this environment (i.e. maximum entropy, pure noise), so learning rates associated with these trials should diminish. Unless outcomes of the declined gamble choices are also used for generating prediction errors (this information should be included explicitly, and again this condition is also 50%), the current learning rates are estimated in a biased fashion relying more to the trials from the higher deck, followed by the even deck and little bit from the low deck as participants usually opt out from this condition. This approach is rather unspecific. I am wondering whether it would be possible to estimate learning rates for each type of deck, similar to modelling Pavlovian Instrumental Learning Task as in that task the conditions are also interleaved. Either way, the authors need to provide a much-detailed description of such confounding factors that I highlighted above. Currently, the paper is lacking important details surrounding these issues.

The reviewer is right that subjects choose to gamble more often with the high as compared to the low deck (cf. Fig. 1C/D), and this means the estimated learning rates reflect subjects’ learning about higher decks to a greater extent. However, it is not correct to think

that an even deck merits a low learning rate, since a subject can only know that a deck is approximately even based upon learning from a substantial number of outcomes. Optimally, learning rates should diminish with the number of observed outcomes. However, the task is short and does not include any experimental modulation of, e.g., volatility, as in previous studies, where it makes sense to allow for changes in learning rates (e.g., Behrens et al., 2007, *Nat Neurosci*; Browning et al., 2015, *Nat Neurosci*). However, we agree with the reviewer, a reasonable alternative is to model distinct learning rates for different decks.

We have now computed a model that incorporates this very aspect. Indeed, a model with three learning rates fits better than a simple Q-learning model with only one learning rate. However, it fits substantially worse (combined BIC across both sessions=16394) than a model with two learning rates that learns differentially from distinct outcomes (combined BIC=16070, with lower BIC scores indicating better fit). Moreover, the three learning rates are not all distinguishable, i.e., parameter estimates are not significantly different from each other. Note that a model with a total of six different learning rates (one per outcome type, positive and negative, per deck) could not be estimated. This was due to the fact that the number of outcomes subjects observed varied substantially across decks and outcome type, such that not all subjects observed both positive and negative outcomes for each one of the decks. For example, on 18 instances, a subject only observed ≤ 1 outcome for one combination of outcome type and deck. Thus, six learning rates cannot be estimated from this data. We believe that a novel design with a larger number of trials would be required to properly test such a model.

Additionally, it is important to note that there is nothing to be learnt from a declined gamble, as outcomes are not shown to participants. We apologise for this apparently confusing task description. We trust we now provide greater clarity in the revised manuscript:

“Subjects were informed that an unsuccessful gamble would result in a loss (-£5), and a successful gamble would result in a win (+£5). Subjects learnt through trial and error about each of the decks’ success likelihood. Alternatively, subjects could decline a gamble and instead opt for a fixed 50% known probability of winning or losing, respectively. After a decision to decline a gamble, the outcome was not shown to participants.” (page 4, results).

“If subjects opted to decline the gamble, they won/lost with a fixed 50% known probability. On such trials, the outcome was not shown to participants.” (page 15, methods)

8) considering the current task involves participants making betting decisions, I am wondering why the authors did not consider risk and loss aversion in their models? I think there needs to be a clear justification as to why the task should be modelled in terms of reinforcement learning rather than risky decision-making, or ideally both. The introduction probably should mention the literature involving serotonergic agents acting on risk seeking tendencies.

We chose a reinforcement learning model for two main reasons. First, to our knowledge, risky decision-making models are typically used in experiments in which the decision variables are presented to subjects explicitly (e.g., Sokol-Hessner et al., 2009, *PNAS*; Rutledge et al., 2014, *PNAS*), whereas our task involves decision variables that need to be learned by trial and error. Second, gains and losses in our experiment have only one possible size. Thus, modelling an entire utility function would not be useful and, in fact, such a function cannot be estimated. What could potentially be derived from a risky decision-making model is the principle that losses may have greater subjective value than similarly

sized gains. However, we tested this possibility (cf. comment #3.5), and the effects in our data seem to be better explained by asymmetric learning, which has been shown to capture risk aversion adequately in the context of trial-and-error learning (Niv et al., 2012, *J Neurosci*; Eldar et al., 2016, *PNAS*). Thus, to address the reviewer's comment, we now discuss our results in light of a literature involving the impact of serotonergic agents on risk-seeking in humans, and we clarify our modelling decisions with respect to risky decision-making literature:

“Note that although our task involves gambling, it contrasts with typical risky decision-making paradigms (e.g., Sokol-Hessner et al., 2009; Rutledge et al., 2014) in that decision variables need to be learned by trial and error. Moreover, the typical approach in risky decision-making studies for estimating a utility function is not suitable here, since gains and losses have only one possible size. Thus, we modelled the data using a variety of reinforcement learning models, which have been shown previously to adequately capture risk sensitivity in the context of trial-and-error learning (Niv et al., 2012, J Neurosci; Eldar et al., 2016, PNAS).” (page 17, methods)

“A learning asymmetry, involving a greater impact of losses than wins, can lead to increased avoidance relative to approach behaviour. This can result in an aversion to risk-taking and action over time, and potentially maladaptive risk-avoidant behaviour (Hauser et al., 2016). However, studies specifically assessing the impact of serotonin on human risk-taking have, thus far, proven inconclusive (Campbell-Meiklejohn et al, 2011; Macoveanu et al., 2013, 2014; Faulkner & Deakin, 2014). Notably, in these studies, decision variables are typically not learned by trial and error but are instead presented to subjects explicitly, which contrasts with the learning design utilised in the current study. Thus, the relationship between a serotonergic effect on asymmetric learning and the development of risk tendencies remains a question for pursuit in future studies.

A learning asymmetry, involving a greater impact of losses than wins, can lead to increased avoidance relative to approach behaviour. This can result in an aversion to taking risks and actions over time, and potentially maladaptive risk-avoidant behaviour (Hauser et al., 2016). However, studies specifically assessing the impact of serotonin on human risk-taking have, thus far, proven inconclusive (Campbell-Meiklejohn et al, 2011; Macoveanu et al., 2013, 2014; Faulkner & Deakin, 2014). Notably, in these studies, decision variables are typically not learned by trial and error but are instead presented to subjects explicitly, which contrasts with the design of the current study. Thus, the relationship between a serotonergic effect on asymmetric learning and the development of risk tendencies remains a question for pursuit in future studies.” (page 11/12, discussion)

Addressing these issues may substantially change the authors results and interpretations in the discussion section. I would be happy to review a revised version of this manuscript in due course. I hope the authors will take these recommendations constructively.

Reviewer #3 (Remarks to the Author):

Michely and colleagues performed a double-blind placebo-controlled study in sixty-six

healthy participants to investigate whether modulating central 5HT-levels affects how much learning is driven by reward and punishment. Half of the participants were administered the SSRI citalopram for seven consecutive days, the other half were given placebo, and all participants performed a gambling task on day 1 and 7. Compared to placebo, citalopram enhanced learning from negative feedback (monetary losses) and diminished learning from positive feedback (monetary gains) on day 7.

This is a nice piece of work which was well designed and thoroughly conducted and analysed. It provides a simple and clear message that is communicated well. I have a few issues with the analyses and interpretation that should be addressed.

We thank the reviewer for a positive evaluation of our study and the constructive suggestions, which we have addressed thoroughly as described below.

Major:

1) The authors are framing their results in a way that might explain why SSRI treatment effects evolve slowly. When framed in terms of the effect asymmetric learning rates have on expectations, the direction of their effects (enhanced learning from negative, diminished learning from positive outcomes) seems intuitive: when expectations are increasingly negative, experiences, by comparison, are overall more positive than expected. However, it becomes more difficult to try and reconcile the results of the present study with the wider literature. For example, there is a body of work showing negative biases are reduced following SSRI treatment. Also, it is well established that healthy people have an optimism bias, they learn more from positive information relative to negative information. However, following the author's logic, healthy people should become depressed because with higher expectations, experiences should be generally disappointing. Furthermore, the optimism bias is not present in depressed patients (Garrett, ... Dolan, 2015; Korn ... Dolan 2013), where learning from negative and positive feedback is more balanced. The results put forward here suggest that SSRIs frequently used to treat depression do not normalize these learning asymmetries in the direction that would move patients towards the healthy population (increasing learning from positive outcomes), but instead push them further away from the healthy population. In other words, negative learning biases thought to contribute to depression are pronounced further after seven days on SSRIs, but the authors argue this could help patients' mood in the long run. This seems counterintuitive and needs further explanation.

We are pleased about the reviewer's comment as it is one we have thought about. We agree we did not discuss this sufficiently in our initial manuscript. We agree it is critical to discuss if, and how, serotonergic effects might be of relevance to understanding the mood-enhancing effects of SSRIs. In this respect, we note that there is an important difference between our approach and the approach commonly taken in the literature on optimism bias. In the latter literature, the assayed learning typically involves updating expectations about oneself given information about the average person (cf. Korn et al., 2014, *Psychol Med*; Garrett et al., 2014, *Front Hum Neurosci*). This type of assay is interesting and important, but it involves an additional critical factor – the degree to which subjects accept that information about the average applies to themselves. There are various reasons to discount the relevance of average stats, and such discounting may be applied asymmetrically for positive and negative statistics. We propose this factor, rather than differences in learning per se, is likely to be a main driver of depression-related individual differences in the optimism bias.

In contrast, our approach strives to assess the basic process of learning from reward and punishment. For this purpose, we designed the study such that feedback from which subjects learn is unequivocal, as typically done in the reinforcement learning literature. Importantly, this and other literature provide substantial evidence that emotional states are strongly affected by prediction errors, which implies that decreasing expectations might in fact lead to greater positive surprise, and as a consequence lead (as has been shown) to a more positive mood (Rutledge et al., 2014, *PNAS*; Eldar et al., 2015, *Nat Commun*; Eldar et al., 2016, *TICS*; Otto et al., 2018, *PLoS One*; Vinckier et al., 2018, *Nat Commun*; Villano et al., 2020, *J Exp Psychol Gen*). This hypothesis fits remarkably well with the present study's finding that SSRIs enhance learning from punishment, and decrease learning from reward.

Of course, in the current study, we are not in a position to relate the impact of changes in learning to changes in mood. Ultimately, in future studies, both hypotheses can be tested in parallel. Ideally, this would include a sample of depressed individuals alongside tracking of subjective changes in mood over time. Such a study could clarify how the two effects relate to one another as well as to changes in mood.

We have now modified the discussion to clarify our perspective on the results in relation to the literature on optimism bias. We hope the reviewer now finds our interpretation, along with its limitations, clearer:

“A related line of work shows that healthy individuals learn more from positive, relative to negative, information leading to an ‘optimism bias’ (Sharot et al., 2007, 2012), where the latter is lacking in depressed patients (Korn et al., 2014; Garrett et al., 2014). This might seem to suggest that an increased learning from positive information acts to protect against depression. It is worth highlighting, however, that these studies typically assay updating expectations about oneself given information about an average person. This type of assay is important, but it involves an additional critical factor – the degree to which subjects accept that information about the average applies to themselves. There are a range of reasons to discount the relevance of average statistics, and such discounting can be applied asymmetrically to positive and negative information. We propose this factor, rather than differences in learning per se, might explain depression-related individual differences in optimism bias. In contrast, our approach strives to assess a more basic process of learning from reward and punishment. For this we designed our study such that the feedback from which subjects learn is unequivocal, as is typically applied in the basic reinforcement learning literature (Niv et al., 2012; Gershman, 2015; Eldar et al., 2016).

Overall, we consider that our results, alongside its putative impact on changes in mood, do not contradict findings on optimism bias in depression. On the contrary, both processes can be expected to contribute to an emergence of better mood (Eldar et al., 2016). Here, a limitation of our study is its restriction to non-depressed healthy individuals. Furthermore, our task did not contain a concurrent mood measurement. Ultimately, testing both self-referential and basic reinforcement learning, alongside tracking of subjective changes in mood in clinical cohorts, will be an important next step for examining a temporal evolution of changes in learning and the emergence of significant clinical effects over time.” (page 12/13, discussion)

2) I have a few comments related to the logistic regression of the choice data (Figure 2) and its relationship to the modelling results (Figure 3).

a. Despite high correlations between learning asymmetries captured with the logistic regression and the two learning rates from the RL model (Figure 3 - supplement 2), it

seems that these two analyses are not necessarily measuring the same behaviour. In an RL model, the strongest weight is given to the most recent outcomes but the cumulative losses regressor does not contain any recency weighting. To make the comparison clearer, the logistic regression could be estimated using recency-weighted cumulative gains/losses, or even simply include the last outcome for the current deck as an additional regressor. It seems that the reason why the comparison looks strong here (Fig 3 - S2, high correlations) is because learning rates are extremely low (Figure 3), so the recency-weighting is relatively flat. Nevertheless, it cannot be assumed that in general, the logistic regression's cumulative regressor and the RL model's learning rate measure the same behavioural influence.

We agree with the referee that the regression and RL model do not precisely mirror one another. However, we deliberately made this choice such that the analyses complement one another. Otherwise, we suspect having both analyses would be redundant. The regression analysis takes a model-agnostic approach, and as the referee rightly infers, uses cumulative predictors that give equal weight to all observed outcomes. This is a sensible analysis because optimal inference requires giving equal weight to all observed outcomes. Showing that this regression analysis and the modelling analysis both yield similar conclusions serves to reassure that our results are robust to changes in analytical strategy.

However, we have followed the reviewer's suggestion and conducted an additional analysis, computing a regression that includes only the last outcome for the current deck as an additional regressor. This regression model is a worse fit (mean model deviance, session I: 100.64, session II: 97.10) than the model presented in our original manuscript (mean model deviance, session I: 95.20, session II: 93.06, with lower deviance indicating better fit), indicating that recency did not play a major role in explaining subjects' choices.

We have now revised the manuscript clarifying precisely how the regression diverges from the computational model, and what is the purpose of each analysis:

“Next, we used a trial-by-trial logistic regression approach (cf. Materials and Methods) to assess whether subjects' decisions to gamble were dependent upon previous receipt of positive, or negative, outcomes over time. First, we found that subjects, over both sessions, gambled more against lower computer numbers (session I: $t_{64}=14.7$, $p<0.001$; session II: $t_{65}=20.9$, $p<0.001$). Second, participants, over the course of each session, gambled more with decks with which they had experienced more success (session I: $t_{64}=10.3$, $p<0.001$; session II: $t_{65}=15.7$, $p<0.001$), and less with decks with which they had experienced more failure (session I: $t_{64}=10.2$, $p<0.001$; session II: $t_{65}=11.0$, $p<0.001$). This result indicates subjects successfully learned about the decks from the outcomes of their gambles.

On session I, effects were similar across drug groups for cumulative success and failure (drug x valence: $F_{1,63}=0.3$, $p=0.844$; drug: $F_{1,63}=0.9$, $p=0.345$; Fig. 2A). At session II, however, we found evidence for an asymmetric impact of outcome success and failure, as a function of treatment (drug x valence, $F_{1,64}=10.5$, $p=0.0018$; drug: $F_{1,64}=2.4$, $p=0.126$; Fig. 2B), attributable to an enhanced impact of failure ($t_{64}=2.3$, $p=0.024$) but not of success ($t_{64}=0.1$, $p=0.892$), in SSRI treated as compared to placebo subjects. This differential pattern suggests that SSRI treatment increased an impact of negative outcomes, enhancing a gamble avoidance tendency in response to failure.”

Next, we used computational modelling (cf. Materials and Methods) to assess the precise learning mechanism underlying the asymmetric effects of success and failure evident in the regression analysis. [...]“ (page 5-7, results)

“Additionally, an asymmetric effect of cumulative success and failure on gambling, as derived from the logistic regression, correlated significantly with an asymmetry in learning, derived from the computational model (session I: $r=0.850$, $p<0.001$, session II: $r=0.841$, $p<0.001$; Fig. 4). Thus, these analyses jointly indicate that altered gambling preference, reflecting asymmetric learning from positive and negative outcomes, was a consequence of serotonergic intervention.” (page 8, results)

b. Relatedly, "cumulative wins" is a somewhat misleading term given it is not merely measuring the cumulative wins (or losses). Wins (losses) are multiplied by the number shown on the computer's deck, so the resulting regressors rather capture how lucky or unlucky the participant has been. (Note, the final RL model also contains this term in the prediction error.)

The reviewer is right that the regressor additionally captures an impact of surprise that is associated with a win, or loss, respectively. In other words, the regressor quantifies the evidence provided by different wins and losses. A win against a high computer number, for instance 7, provides more evidence that the deck with which one played contains high numbers. Specifically, in this case it would mean that the card drawn from the deck was either an 8 or 9. By contrast, winning against a low number, for instance 2, would only mean that we drew a number that is 3 or greater. Thus, the latter win does not provide as much evidence in favour of our deck. Similarly, losing against a low number provides more evidence against our deck compared to losing against a high number. This computation, as the reviewer rightly points out, mirrors the computation of prediction errors in the winning computational model, the adjusted Q-learning model. We have now clarified this point in the revised version of the manuscript, explaining the rationale for referring to this term as cumulative “success/failure”, instead of merely cumulative “wins/losses”.

“ x_{2_t} represents cumulative success, reflecting, for the deck played with on trial t , the sum of previous positive outcomes, computed as $+1$ multiplied by the computer's number against which it was received. x_{3_t} represents cumulative failure, reflecting, for the deck played with on trial t , the sum of previous negative outcomes, computed as -1 multiplied by $(10 - \text{computer's number})$, against which it was received. The multiplications by the computer's number reflect the fact that a win against a higher computer number provides stronger evidence in favour of a deck, and similarly, a loss against a lower computer number provides stronger evidence against a deck. Thus, we refer to the regressors as cumulative “success/failure”, instead of merely cumulative “wins/losses”. To adequately compare effect sizes between coefficients, x_{2_t} and x_{3_t} were range-normalized between -1 and 1 .” (page 17, methods)

c. The logistic regression should include interaction effects. For example, one might expect an interaction between computer's drawn number and deck type. It would also be interesting to examine whether losses in general versus losses received for the present card deck (i.e. respecting correct credit assignment, as currently coded, or not) are driving the asymmetries seen following citalopram administration. This could be distinguished by comparing two models, one containing cumulative wins/losses for the current card deck, the other containing cumulative wins/losses across all card decks.

We followed the reviewer's suggestion and computed a control analysis, assessing an interaction term between computer number and deck type. Critically, on both sessions, the

regressor is not significant at the overall population level, i.e., there is no evidence for a significant interaction (session I: $p=0.313$; session II: $p=0.077$). Based upon a suggestion by reviewer 2 (cf. #2.5), however, the regressor deck type was removed from our regression analysis altogether, thus rendering this issue less relevant for our novel analysis.

The reviewer raises another interesting issue regarding a second regression that contains cumulative success/failure across all card decks, rather than for the current card deck. We have now computed this analysis. However, such a regression model fits worse (mean model deviance, session I: 100.93, session II: 97.02) than the model presented in our original manuscript (mean model deviance, session I: 95.20, session II: 93.06, with lower deviance indicating better fit). We thus decided to stick to our current analysis, and we hope that the reviewer agrees with this.

3) The full interaction session x drug x valence is not significant. To conclude that the drug effect was specific to day 7, however, it would be necessary to show an interaction with session which is currently missing. I wondered whether the session x drug interaction is significant for either positive or negative learning rates (or both) in Figures 2 + 3?

We agree with the reviewer that to conclude that the drug effect was specific to day 7, it would be necessary to show a three-way interaction. We have now computed an interaction for positive and negative learning rates (drug x session) but did not find a significant interaction for either (positive: $p=0.206$; negative: $p=0.255$). However, although we cannot conclude that the effect of the drug is stronger on day 7 as compared to day 1, the significant drug x valence interaction that we report shows that there is a drug effect overall. In light of the reviewer's comment, we have toned down a session-specific interpretation of the findings, and now discuss ways future study designs can overcome this limitation:

“Although our data suggests an emergence of serotonergic effects after a temporally extended intervention, a three-way interaction (drug x session x valence) was not significant. Thus, we can conclude the drug had an hypothesized effect, but not that this effect was significantly greater after prolonged (week-long, on day 7) as compared to acute (single-dose, day 1) treatment. To unravel the precise trajectory of any such effect, future studies will ideally include a pre-drug testing session as well as multiple sessions over several weeks of treatment.” (page 13, discussion)

4) Have the authors tried a model with a positive and negative learning rate per card deck (a total of six learning rates)? I don't know if such a model would be estimable given the number of trials, but this can be checked using simulations. This is interesting because the distribution of the numbers drawn by different decks differs, and this might affect their respective uncertainty/volatility estimate, which we know from Pulcu & Browning's work affects the learning rate.

We agree an interesting option is to model distinct learning rates for different decks. We have now computed a model that incorporates this suggestion. Indeed, such a model with three learning rates fits better than a simple Q-learning model with only one learning rate. However, it fits substantially worse (BIC across both sessions=16391) than a model with two learning rates that learns differently from distinct outcomes (BIC=16070). Moreover, the three learning rates are not all distinguishable, i.e., parameter estimates are not significantly different from each other. However, a model with a total of six different learning rates (one per outcome type, positive and negative, per deck) could not be estimated due to the fact that

the number of outcomes subjects observed varied substantially across decks and outcome type, such that not all subjects observed both positive and negative outcomes for each one of the decks. For instance, on 18 instances, a subject only observed ≤ 1 outcome for a combination of outcome type and deck. Thus, six learning rates cannot be estimated from this data. We believe that a novel design with a larger number of trials is required to properly test such a model.

5) The parameter recovery is either wrong or its description is inaccurate. The standard is to produce choices from an artificial agent with a chosen set of parameters and then plot those chosen parameters ('ground truth') against the parameters fitted to the choices produced by this artificial agent ('recovered parameters'). This is then repeated by systematically varying all parameters in the expected range. However, Suppl Fig 4 seems to show a relationship between simulated and real data, rather than between ground truth and recovered (both relating to the synthetic data). This is either a reporting mistake or a more fundamental error, but establishing reliable parameter recovery is critical.

We thank the reviewer for this comment. We provided evidence for the accuracy of our computational modelling approach in a variety of ways. Firstly, we demonstrated that generating simulated data based upon model parameters derived from fitting to real data shows that the model captured core features of the real data (Supplementary Fig. S3), an approach referred to as 'posterior predictive check'. Secondly, we simulated data with different models and demonstrated that our model comparison procedure could accurately identify the model that generated the data (i.e., 'model recovery test'; Supplementary Fig. S5). Thirdly, we demonstrated that parameter estimates could be accurately recovered (i.e., 'parameter recovery test'; Supplementary Fig. S4).

We understand that the reviewer's comment is related to the latter aspect, which we would like to clarify. In fact, for Supplementary Fig. S4, as suggested by the reviewer, we had produced choices from an artificial agent with a chosen set of parameters and plotted these parameters ('ground truth') against the parameters fitted to the choices produced by this artificial agent ('recovered parameters'). Thus, Supplementary Fig. S4 indeed shows a relationship between parameters that both relate to synthetic data, not to real data. The confusion seems to have arisen because the range of parameter settings ('ground truth') used to generate the data was chosen so as to cover the empirical range (the range derived from fitting to real data).

To clarify this, we have now revised the relevant text in the methods and results section:

"We performed several analyses to assess the validity of our computational modelling approach. First, we generated simulated data based upon model parameters derived from fitting to real data. This 'posterior predictive check' confirmed that the model captured core features of the real data (Supplementary Fig. S3). Additionally, we simulated sets of choices from artificial agents with specific sets of parameters ('ground truth') and then fitted models to those choices to recover the values of the parameters ('recovered parameters'). To ensure the results of the parameter recovery test were applicable to the analysis of the real data, we selected the ground truth parameters such that they covered the empirical range. This analysis revealed that individual parameter estimates could be accurately recovered (Supplementary Fig. S4). Lastly, we validated our model comparison procedure by generating simulated data using each model and applying our model comparison procedure

to identify the model that generated each dataset (Supplementary Fig. S5).” (page 9/10, Results)

Minor Comments:

1) What was the rationale for completing the first behavioural session after a single dose of citalopram rather than before any citalopram administration?

The rationale for completing two sessions, on different days of treatment, was to be in a position to assess the effects of both acute (single-dose) and prolonged (week-long)

Figure 2. Results of trial-by-trial logistic regression model.

Fitting a logistic regression model to subjects’ decisions showed that participants gambled more against lower computer numbers, with no drug differences on session I (A), and session II (B), respectively. Additionally, subjects, over the course of a session, gambled more with increasing success with each deck, and gambled less with increasing failure with each deck. On session I, impact of cumulative success and failure was unaffected by treatment (A). On session II, however, SSRIs enhanced the impact of failure but not wins (B), indicating an asymmetric drug effect on reward and punishment. **p<0.01, *p<0.05, n.s.=not significant. Error bars indicate SEM.

treatment. Certainly, adding a baseline (pre-drug) session could have been beneficial. In this study, however, we were not in a position to do so for practical reasons. We now add additional information regarding the study design to the revised manuscript:

“Thus, subjects were assessed twice, once after single-dose, and once after week-long administration of the drug. This repeated-measures study design enabled (i) an assessment of a pharmacological effect overall, as both sessions were performed under the influence of the drug, and (ii) an assessment of putative differences between acute (single-dose) and prolonged (week-long) treatment.” (page 14/15, methods)

2) Is it the case that the first two regressors are scaled between -1 and 1 in Figure 2 (logistic regression) and the next two (cumulative wins/losses) between 1-9? This won't affect any conclusions, but it is standard to range-normalize or z-score regressors to make effect sizes comparable.

We thank the reviewer for this constructive comment. We have range-normalized the two regressors, in line with the other regressors. As expected, statistics and conclusions remain unchanged, but effect sizes are now comparable. See below for a revised version of Fig. 2.

3) The dopamine-serotonin opponency theory is surprisingly prominent given it is no longer considered timely.

We have removed an explicit reference to the opponency theory, whilst maintaining crucial references regarding the involvement of dopamine and serotonin in reward and punishment learning. We hope the reviewer finds our introduction and discussion section more suitable in its revised form.

“Previous research shows that the neuromodulators dopamine and serotonin play a key role in modulating reward and punishment learning. Whilst there is ample evidence for a role of dopamine in learning from reward¹⁸⁻²¹, the evidence in relation to serotonin is less clear. Some studies indicate a specific role in punishment learning²²⁻²⁶, while others report that serotonin impacts learning from both reward and punishment^{27,28}.” (page 2, introduction)

“Serotonin is an evolutionary conserved neurotransmitter though its precise effects on cognition have evaded a definite mechanistic understanding^{40,41}. One influential proposal is that serotonin plays a specific role in processing aversive outcomes⁴². Indeed, several studies in humans show that serotonin is involved in punishment learning²²⁻²⁶, but other studies suggest that it impacts learning from both positive and negative outcomes^{27,28}.” (page 9, discussion)

4) Please report the optimal learning rate in this task. Intuitively, it seems that slower learning is better than faster learning given the card decks have a constant number distribution. This could change the interpretation of learning rate changes seen in Figure 3.

The reviewer raises an interesting issue here, and we have now computed a number of control analyses to test this. Specifically, we simulated data (10 data sets each) from 19 models, each of which with a different set of learning rates, for both sessions. We varied learning rates from 0.01-0.09, and 0.1-1, respectively. Subsequently, we computed the net reward won by artificial agents with these settings, on average across 10 simulated data sets (shown in blue for 0.01-0.09, and black bars for 0.1-1 below). We also compared it with the net reward won using learning rates derived from fitting to real data (green bar), as well as to the net reward subjects received in the real experiment (red bar).

This analysis shows that learning rates in the range of $\approx 0.1-0.7$ reap the highest reward, with an ‘optimal’ learning rate in the range of $\approx 0.3-0.6$, respectively. The results also indicate that lower (e.g., ≤ 0.05), and higher (e.g., ≥ 0.8), learning rates are detrimental to performance. Overall, in our study, the mean learning rate, at the population level, was 0.125 (green bar), which allowed subjects, in the real experiment, to reap a net reward (red bar) that was close to, but still lower than, the net reward of artificial agents with an ‘optimal’ learning rate.

Additionally, we found that drug groups did not differ net reward (session I: $p=0.200$; session II: $p=0.649$). Thus, we conclude that faster learning in our task is advantageous for performance. Note that, however, SSRI treatment increased punishment, but decreased reward learning, thus modified an asymmetry in learning, which did not impair or improve task performance per se. We have added this information to the revised manuscript as follows:

“We found no difference between drug groups in net reward gained, a key measure for task performance (session I: $p=0.200$; session II: $p=0.649$). Thus, there was no evidence that changes in asymmetric learning were detrimental, or advantageous, for task performance.” (page 9, results)

“Note that data simulation showed that an ‘optimal’ learning rate for the task, maximizing net reward gained, was in the range of $\approx 0.3-0.6$.” (page 7, legend Fig. 3)

5) Did the authors test a model with outcome sensitivity parameters (for gains and losses)? Could sensitivity to outcomes rather than changes in updating be explaining the results?

We thank the reviewer for this suggestion. We have now computed different models testing for a sensitivity to outcomes and compared it to changes in learning. Across both sessions, a model with different sensitivity to distinct outcomes fits worse (combined BIC=15126) than a model that learns asymmetrically from distinct outcomes (combined BIC=15117). Additionally, we tested a model that modulates both a sensitivity to distinct outcomes, and learns differently from distinct outcomes. Although the latter is clearly not the best-fitting model (combined BIC=15148), we used it to test drug effects on the parameters. Here we found that a sensitivity to outcomes is not different between drug groups ($p=0.582$), but asymmetric learning is modulated by SSRIs ($p=0.0019$), with reduced learning from reward ($p=0.004$), and increased learning from punishment ($p=0.031$) after serotonergic intervention. These drug differences mirror the findings presented in our manuscript. Thus, we believe these control analyses increase confidence in our results. To keep comparability with our prior study using an identical task (Eldar et al., 2016, *PNAS*) and other studies assessing learning asymmetries (e.g., Niv et al., 2012, *J Neurosci*; Gershman, 2015, *Psychon Bull Rev*; Lefebvre et al., 2017, *Nat Hum Behav*), we decided to stick to our current model space and hope the reviewer agrees with our line of reasoning. Additionally, we now describe the above-mentioned control analyses in the revised manuscript:

“Note in addition we tested a model with differing sensitivity to outcome valence (positive and negative, respectively), and a model that modulates both a sensitivity to distinct outcomes and learns differently from these distinct outcomes. Across both sessions, the models provided a worse fit than a model that learns asymmetrically from distinct outcomes. Although the latter model was clearly not the best-fitting model, we used it for a joint test of drug effects on both outcome sensitivity and learning parameters from session II. Here we

found that a sensitivity to outcomes did not differ between drug groups ($t_{64}=0.5$, $p=0.582$), but that asymmetric learning was modulated by SSRIs (drug \times valence: $F_{1,64}=10.4$, $p=0.0019$), with reduced learning from reward ($t_{64}=2.9$, $p=0.004$), and an increased learning from punishment ($t_{64}=2.1$, $p=0.032$) following serotonergic intervention, mirroring the drug effects on parameters of our winning model.” (page 9, results)

6) Please add individual data points to figures.

7) Please report precise p-values, throughout.

8) Figure 2 is missing labels A-D.

We apologize for these omissions and have now incorporated all the requested changes in 6) – 8).

Reviewers' comments:

Reviewer #1 (Remarks to the Author):

The authors successfully addressed my concerns, the paper is much improved and I recommend publication of the paper.

Reviewer #2 (Remarks to the Author):

Review of revision by Michely et al 2021

I would like to thank the authors for doing detailed analysis and their responses on my previous recommendations. Although majority of the key issues have been addressed, the paper still requires further revisions before acceptance for publication. Again, I would hope that the authors will take these recommendations constructively.

First of all, I think the authors need to report the overarching ANOVA results (from line 116) at the beginning of their logistic regression analysis rather than at the very end. If we take a few steps back, the paper is asking 3 questions:

does a single dose of citalopram influence learning? No

does a prolonged dosing regimen influence learning? Yes

does prolonged dosing influence learning over and above a single dosage? No.

All of these messages above should be included in the abstract, along with the finding that serotonin does not lead to any detrimental or advantageous outcomes.

In my initial review, I had claimed that majority of the significance between drug and placebo groups arising on the 7th day (according to the logistic regression analysis) is due to the fact that the placebo group behaves differently (figure 2B). I still think majority of the readers will come to this conclusion.

In authors rebuttal, they suggest that my interpretation is inconsistent with the evidence from learning rates. The authors suggest that the learning does not change significantly in the placebo group and further compare placebo and serotonin groups based on the level of statistical significance ($p < .001$ versus $p = .054$). Unfortunately, in medical sciences comparisons between groups should not be based on how they relate to a baseline but should be compared directly. The 2nd important issue to point out here is, if the result of a statistical test is in a hypothesized direction even higher p-values are descriptively reported as approaching significance etc., whereas in the case illustrated above, even a slight deviation from .05 is regarded as nonsignificant. This language needs to be unified throughout the paper, I think p-values over .1 should be regarded as non-significant, whereas p-values above .065 can be regarded as approaching significance and values between .05-.065 can be described as marginally significant. All of these thresholds are arbitrary, the authors can refer to an independent scientific source, but the important thing is they need to be consistent throughout the paper.

In my opinion, the results suggesting a significant effect of serotonin identified by computational modelling is still important. However, the authors need to provide some explanation as to why they were able to isolate these differences with computational modelling, which is not apparent in the logistic regression model. What kind of variance does computational modelling account for, which is not captured by the logistic regression model. In figure 4, the authors show the correlation between logistic regression coefficients and learning rates, which may shed some light into this question. However, these correlations are computed across the whole cohort. When there are between group differences, this approach can inflate the degrees of freedom and may yield significant results solely based on differences between 2 groups, can also lead to what is referred to as Simpson's paradox. The authors need to compute these correlations separately for each group and for each session, and figure 4 should be presented as a 4-panel figure. This is an extremely important issue.

I might have a few smaller questions:

how were participants reimbursed? What was the relationship between financial outcomes in the

learning sessions and final reimbursement?

The authors report on BIC scores for model selection, which is perfectly okay. I believe they also need to report the average predictive accuracy of this model (i.e., percent correct). This information should also be included in the abstract and reported as a histogram as a supplementary figure.

Another thing that I do not agree from the previous revision is, the assertion that separate learning rates are not needed because the task does not have any volatility manipulation. The learning rates also change as a function of the strength of the probabilistic association, aka expected uncertainty (e.g., Yu and Dayan or more recent work from Daw's lab). The fact that task design does not allow further dissection of participant learning behaviour, the fact that learning rates are modelled across the conditions with more trials coming from the high deck followed by mid and low deck condition should be written in the limitations.

Finally, figure 1D suggests that groups were particularly different on the 50-50 deck on day 7. An interpretation as to why this may be the case would be interesting to know.

Looking forward to seeing the final version of this paper.

Reviewer #3 (Remarks to the Author):

The authors have thoroughly addressed my previous comments and added several helpful analyses. The manuscript has become clearer and more convincing as a result. I have a few outstanding issues, all of them except the first are very minor:

1. Two reviewers (R2/R3) highlighted the disconnect between the model-free and model-based analyses. The text changes implemented in the manuscript in response to this comment do not sufficiently address this question. Previous outcomes are weighted equally in the logistic regression analysis but are recency-weighted (dependent on the learning rate) in the computational modelling analysis and it is not clear which of the two better explains behaviour. In particular, it is unclear whether the correlation reported in Figure 4 is simply due to similar constructs being measured (in which case the plot is a good sanity check but maybe better placed in the Supplement), or indeed an interesting result that adds further robustness to the conclusions, which is how this result is portrayed at the moment. This can be tested by simulating agents with different positive/negative learning rates (e.g., varying both independently between 0.1-0.8) and establishing the correlations between the two types of analyses in these artificial datasets. I feel this is important to fully establish if the correlation adds value to the manuscript or is a result of the analysis pipeline. I would not insist on this point if the two results in Fig2 and Fig3 were not the core message of this manuscript. It seems crucial that readers can immediately understand whether and how these two types of analyses relate conceptually and methodologically.

2. I think other readers will be interested in the additional models that have been tested in response to the reviewer's comments and included in the rebuttal, but these are currently not part of the revised manuscript. It would be good to mention them at least briefly in the supplement, maybe in the text or figure legends (e.g., related to points 2c and 4: three learning rates for different decks – worse fit; six learning rates - not estimable; deck-unspecific cumulative failure/success – worse fit).

3. Similarly, given the authors have put effort into this analysis, I can see no disadvantage in including the simulation of the optimal learning rate as a supplementary figure (response to previous point 4). It is now mentioned briefly in the legend of Fig3, but readers might wonder how this was established.

4. Related to my previous point 5 (parameter recovery), I think the confusion was purely down to the x and y-axis labels in Figure S3. It is somewhat confusing to refer to x as 'fitted to real data' and y as 'fitted to simulated data'. I suggest changing these labels to something a bit more intuitive and which conveys both axes relate to the artificial data (e.g., "simulated parameter (chosen based on parameters in real data)" and "recovered parameter" or similar)

5. p-values are still indicated as $p < 0.001$ etc in the results - please report precise p-values.

6. Statements relating to non-significant effects (e.g., Effects were similar across drug groups") are sometimes framed as if they provide support for the null, but this would require Bayesian statistics (frequentist statistics cannot provide evidence for the null). Bayesian model evidence for the null model (compared to alternatives) are very easy to compute using free software such as JASP. Alternatively, the statements should be reframed as not supporting a difference, rather than as evidence for the null, e.g. "There was no evidence for a difference between drug groups."

7. In Figure3, in the individual data points, it is noticeable that the between-subject variance is much smaller for negative outcomes in session II both for the SSRI and placebo group. Do the authors have any thoughts on why this might be?

We thank the referees for the constructive review process, and their positive evaluation of our revised manuscript “Serotonin modulates asymmetric learning from reward and punishment” (COMMSBIO-20-3363A). We addressed the remaining concerns below. The respective changes in the manuscript are marked in bold. Please note that the page numbers given below refer to the revised manuscript.

Reviewer #1 (Remarks to the Author):

The authors successfully addressed my concerns, the paper is much improved and I recommend publication of the paper.

We thank the reviewer for the appreciation of our work and their previous suggestions that helped us to substantially improve the manuscript.

Reviewer #2 (Remarks to the Author):

Review of revision by Michely et al 2021

I would like to thank the authors for doing detailed analysis and their responses on my previous recommendations. Although majority of the key issues have been addressed, the paper still requires further revisions before acceptance for publication. Again, I would hope that the authors will take these recommendations constructively.

We are grateful to the reviewer for the constructive review process that led to an improved version of our manuscript. We have addressed the remaining concerns as described below.

**1) First of all, I think the authors need to report the overarching ANOVA results (from line 116) at the beginning of their logistic regression analysis rather than at the very end. If we take a few steps back, the paper is asking 3 questions:
does a single dose of citalopram influence learning? No
does a prolonged dosing regimen influence learning? Yes
does prolonged dosing influence learning over and above a single dosage? No.
All of these messages above should be included in the abstract, along with the finding that serotonin does not lead to any detrimental or advantageous outcomes.**

The reviewer is correct that the main messages should be clear to the reader. We have followed the reviewer’s suggestion and modified the presentation of our results (starting with the overarching 3-way ANOVA), both for the regression and modelling analyses. Additionally, we have substantially revised the abstract so as to render clear the key messages alongside describing that serotonin does not impact overall task performance. Moreover, we mention the lack of a significant 3-way interaction specifically in the limitations section. We have revised the manuscript accordingly.

“Next, we used a trial-by-trial logistic regression approach (cf. Materials and Methods) to assess whether subjects’ decisions to gamble were dependent upon the computer number and previous receipt of positive, or negative, outcomes over time. First, we found that

subjects, over both sessions, gambled more against lower computer numbers (session I: $t_{64}=14.7$, $p=3.0e-22$; session II: $t_{65}=20.9$, $p=1.3e-30$; Fig. 2), with no evidence for a difference between drug groups (drug: $F_{1,63}=0.4$, $p=0.505$; drug \times session: $F_{1,63}=0.06$, $p=0.805$).

Second, participants, over the course of each session, gambled more with decks with which they had experienced more success (session I: $t_{64}=10.3$, $p=3.0e-15$; session II: $t_{65}=15.7$, $p=8.2e-24$) and less failure (session I: $t_{64}=10.2$, $p=4.3e-15$; session II: $t_{65}=11.0$, $p=1.6e-16$). This result indicates subjects successfully learned about the decks from the outcomes of their gambles. When assessing data across both sessions, the pharmacological effect on gambling preferences as a function of outcome type was not statistically significant (drug: $F_{1,63}=1.7$, $p=0.194$, drug \times valence: $F_{1,63}=2.6$, $p=0.108$, drug \times session \times valence: $F_{1,63}=2.4$, $p=0.124$). However, analysing both sessions separately, effects were similar across drug groups for cumulative success and failure on session I (drug \times valence: $F_{1,63}=0.3$, $p=0.844$; drug: $F_{1,63}=0.9$, $p=0.345$; Fig. 2A), whereas on session II we found evidence for an asymmetric impact of success and failure outcomes, as a function of treatment (drug \times valence, $F_{1,64}=10.5$, $p=0.0018$; drug: $F_{1,64}=2.4$, $p=0.126$; Fig. 2B), attributable to an enhanced impact of failure ($t_{64}=2.3$, $p=0.024$) but not of success ($t_{64}=0.1$, $p=0.892$), in SSRI treated as compared to placebo subjects.” (page 6/7, results)

“When assessing computational parameters across data from both sessions, we found a significant asymmetric effect of SSRIs on learning rates (drug \times valence: $F_{1,62}=4.1$, $p=0.046$; drug: $F_{1,62}=0.8$, $p=0.365$), but no significant three-way interaction (drug \times valence \times session: $F_{1,62}=1.0$, $p=0.305$, controlling for an overall gambling bias, Supplementary Fig. S3). Follow-up tests revealed that, on session I, computational parameters governing the rate of learning from reward and punishment were similar across treatment groups (drug \times valence: $F_{1,64}=0.007$, $p=0.933$; drug: $F_{1,64}=0.3$, $p=0.553$; Fig. 3A). However, by session II, a significant serotonergic impact on learning asymmetry was evident (drug \times valence: $F_{1,64}=8.2$, $p=0.006$; drug: $F_{1,64}=3.2$, $p=0.075$; Fig. 3B), such that in SSRI, as compared to placebo subjects, learning from reward was reduced ($t_{64}=2.7$, $p=0.008$) while learning from punishment was enhanced ($t_{64}=2.0$, $p=0.041$).

Overall, these results indicate that a prolonged regimen of SSRI treatment resulted in a modulation of learning asymmetries. Importantly, there were no between-group differences for the remaining model parameters (Supplementary Fig. S3). With regards to the impact of a single SSRI dose, on the one hand there was no significant impact following a single dose, while on the other the impact following a single dose did not significantly differ from the impact following prolonged SSRI treatment.” (page 8/9, results)

“Instrumental learning is driven by a history of outcome success and failure. Here, we examined the impact of serotonin on learning from positive and negative outcomes. Healthy human volunteers were assessed twice, once after acute (single-dose), and once after prolonged (week-long) daily administration of the SSRI citalopram or placebo. Using computational modelling, we show that prolonged boosting of serotonin enhances learning from punishment and reduces learning from reward. This valence-dependent learning asymmetry increases subjects’ tendency to avoid actions as a function of cumulative failure without impacting overall task performance. By contrast, no significant modulation of learning was observed following acute SSRI administration. However, differences between the effects of acute and prolonged administration were not significant. We discuss these findings with respect to how serotonergic agents may impact on mood disorders.” (abstract, page 1)

“Although our data suggests an emergence of serotonergic effects after a temporally extended intervention, it is of note that a three-way interaction (drug x session x valence) was not significant. Thus, we tentatively conclude that prolonged treatment induced a learning asymmetry, but the interpretation of this needs to be tempered by the fact that there was no difference between prolonged (week-long, on day 7) as compared to acute (single-dose, day 1) treatment. To unravel the precise trajectory of any such effect, future studies should ideally include a pre-drug testing session as well as multiple sessions over several weeks of treatment.” (limitations, page 14/15)

2) In my initial review, I had claimed that majority of the significance between drug and placebo groups arising on the 7th day (according to the logistic regression analysis) is due to the fact that the placebo group behaves differently (figure 2B). I still think majority of the readers will come to this conclusion. In authors rebuttal, they suggest that my interpretation is inconsistent with the evidence from learning rates. The authors suggest that the learning does not change significantly in the placebo group and further compare placebo and serotonin groups based on the level of statistical significance ($p < .001$ versus $p = .054$). Unfortunately, in medical sciences comparisons between groups should not be based on how they relate to a baseline but should be compared directly. The 2nd important issue to point out here is, if the result of a statistical test is in a hypothesized direction even higher p-values are descriptively reported as approaching significance etc., whereas in the case illustrated above, even a slight deviation from .05 is regarded as nonsignificant. This language needs to be unified throughout the paper, I think p-values over .1 should be regarded as non-significant, whereas p-values above .065 can be regarded as approaching significance and values between .05-.065 can be described as marginally significant. All of these thresholds are arbitrary, the authors can refer to an independent scientific source, but the important thing is they need to be consistent throughout the paper.

We agree and have now revised the text so as to unify the description of statistical thresholds throughout the manuscript. Additionally, we agree that conclusions should be based on between-group comparisons. We apologize for a potentially misleading response in the previous response letter. By providing the statistics of a within-group comparison (vs. baseline) in the response letter, we aimed to show that the placebo group is in fact *not* showing a substantial difference between session 1 and 2 (as initially suggested by the reviewer). Note, however, this statistic is not reported in the manuscript. In fact, we would emphasise that all results presented in our manuscript rely on between-group comparisons, and we do not draw conclusions based on within-group comparisons.

3) In my opinion, the results suggesting a significant effect of serotonin identified by computational modelling is still important. However, the authors need to provide some explanation as to why they were able to isolate these differences with computational modelling, which is not apparent in the logistic regression model. What kind of variance does computational modelling can account for, which is not captured by the logistic regression model. In figure 4, the authors show the correlation between logistic regression coefficients and learning rates, which may shed some light into this question. However, these correlations are computed across the whole cohort. When there are between group differences, this approach can inflate the degrees of freedom and may yield significant results solely based on differences between 2 groups, can also lead to what is referred to

as Simpsons' paradox. The authors need to compute these correlations separately for each group and for each session, and figure 4 should be presented as a 4-panel figure. This is an extremely important issue.

We thank the referee for these observations.

First, we believe the regression and modelling analyses reveal similar findings (which is mirrored by the significant correlations and the drug effects on session II). At the same time, they do not reflect the exact same analysis of behaviour (which is also why we presented them as separate analyses, as, otherwise, the presentation of the regression results would be redundant).

Second, we now follow the reviewer's suggestion and compute correlations separately, for each group and session, i.e., four separate analyses. The results show that correlations between an asymmetry in regression and modelling results are highly significant across groups and sessions: session I, placebo: $r=0.873$, $p=7.1e-11$, session I, SSRI: $r=0.819$, $p=5.7e-9$; session II, placebo: $r=0.841$, $p=8.7e-10$; session II, SSRI: $r=0.737$, $p=9.8e-7$). This reveals a relationship between an asymmetry in gambling behaviour as a function of outcome valence (regression), and an asymmetry in learning as a function of outcome valence (modelling). We believe this is evidence these correlations are not a result of the group differences, but reflect a conceptual relationship between the analyses.

We now provide the results of these additional analyses alongside a novel 4-panel figure in the revised version of the manuscript. Additionally, we have followed suggestions of reviewer #3 to compute additional simulation analyses, which nicely demonstrate the conceptual relationship between the two measures (cf. #3.1).

“Additionally, an asymmetric effect of cumulative success and failure on gambling, as derived from the logistic regression, correlated significantly with an asymmetry in learning, as derived from the computational reinforcement learning model, in both sessions for both drug groups (session I, placebo: $r=0.873$, $p=7.1e-11$, session I, SSRI: $r=0.819$, $p=5.7e-9$; session II, placebo: $r=0.841$, $p=8.7e-10$; session II, SSRI: $r=0.737$, $p=9.8e-7$; Fig. 4).” (page 9/10, results)

Figure 4. Asymmetric effects of reward and punishment.

An asymmetric effect of cumulative success and failure on gambling, as derived from the logistic regression, significantly correlated with an asymmetry in learning, as derived from our computational model, in both sessions for both drug groups. (A) session I, placebo. (B) session I, SSRI. (C) session II, placebo. (D) session II, SSRI.

I might have a few smaller questions:

4) how were participants reimbursed?

&

5) What was the relationship between financial outcomes in the learning sessions and final reimbursement?

We have now added this information to the revised manuscript.

“Subjects were reimbursed for their time. Additionally, subjects were informed that, at the end of the experiment, one trial was randomly selected, and the outcome of that trial was added to the overall payment. Thus, performance was incentivised as choosing good gambles resulted in a higher probability of earning additional monetary reward” (page 16, methods)

6) The authors report on BIC scores for model selection, which is perfectly okay. I believe they also need to report the average predictive accuracy of this model (i.e., percent correct). This information should also be included in the abstract and reported as a histogram as a supplementary figure.

We now provide predictive accuracy of the winning model, i.e., percent correct, in the revised manuscript and the respective figure in the supplementary material.

“The predictive accuracy of the model (absolute fit), i.e., the proportion of subjects’ choices to which the model gives a likelihood greater than 50% (percent correct) was 87.71% for session I, and 87.92% for session II (Supplementary Fig. S1)” (page 8, results)

7) Another thing that I do not agree from the previous revision is, the assertion that separate learning rates are not needed because the task does not have any volatility manipulation. The learning rates also change as a function of the strength of the probabilistic association, aka expected uncertainty (e.g., Yu and Dayan or more recent work from Daw’s lab). The fact that task design does not allow further dissection of participant learning behaviour, the fact that learning rates are modelled across the conditions with more trials coming from the high deck followed by mid and low deck condition should be written in the limitations. Finally, figure 1D suggests that groups were particularly different on the 50-50 deck on day 7. An interpretation as to why this may be the case would be interesting to know.

We agree with the reviewer that the lack of a volatility manipulation does not imply that learning rates are constant. Optimally, learning rates should diminish with the number of observed outcomes. However, the task was rather short, and the number of outcomes subjects observed varied across decks and outcome type. Thus, for instance, a total of six different learning rates (one per outcome type, positive and negative, per deck) could not be estimated. We agree with the reviewer this should be mentioned as a limitation. We have revised the manuscript accordingly.

“Note that a model with different positive and negative learning rates for each deck could not be estimated due to the number of outcomes subjects observed varying substantially across decks and outcome type, such that not all subjects observed both positive and negative outcomes for each of the decks. Thus, in accordance with our earlier work using an equivalent task (Eldar et. al., 2016), we assumed the same two learning rates characterized learning about all decks. We acknowledge a limitation of this approach is that learning rate estimation is more heavily influenced by trials from the high, followed by the even and then the low deck, as subjects gambled more often with better decks and consequently observed more outcomes from which they could learn.” (page 22, methods)

Addressing the 2nd point raised by the reviewer (re. Fig. 1D): We thank the reviewer for bringing this to our attention. To interpret the depicted result, it is critical to mention that

the experiment was designed to adapt the computer's numbers such that all participants would gamble on approx. 50% of trials. We describe this procedure in the methods section of the manuscript:

“To ensure that all participants gambled in approximately 50% of trials, the numbers that the computer drew three times each were increased by one (e.g., [4, 5, 6] to [5, 6, 7]), in each subsequent set of 15 trials, if subjects took two thirds or more of the gambles against these numbers in the previous 15 trials, or decreased by one if participants took a third or less of the gambles.” (page 18, methods)

Indeed, this adaptation worked, i.e., the overall proportion of accepted gambles was approx. 50%, and there was no evidence for differences between drug groups in either session (cf. results below). Given similar overall gambling frequencies, the question remains as to whether, as subjects learned about the decks, a difference developed in the two groups' relative frequency of gambling for different types of decks. Testing this statistically revealed no significant difference between drug groups (cf. results below). We thank the reviewer for bringing this to our attention and revised the manuscript accordingly.

“The experiment was designed such that the computer numbers changed over time to ensure subjects gambled on approximately 50% of trials across all decks (cf. Materials and Methods). Indeed, this adaptation worked, and overall proportion of accepted gambles did not differ between drug groups (session I: SSRI 49.0%, placebo 49.3%, $t_{65}=0.1$, $p=0.846$; session 2: SSRI 50.6%, placebo 51.2%, $t_{65}=0.4$, $p=0.641$). Thus, evidence of learning manifested in how the rate of accepted gambles differed between decks, and in an observation that this difference grew over the course of the experiment, i.e., from 1st to 3rd block (Fig. 1C/1D; session I: low vs. even, $t_{65}=3.2$, $p=0.0016$; low vs. high, $t_{65}=6.1$, $p=6.0e-8$; even vs. high, $t_{65}=3.0$, $p=0.003$; session II: low vs. even, $t_{65}=2.9$, $p=0.003$; low vs. high, $t_{65}=6.1$, $p=6.2e-8$; even vs. high, $t_{65}=2.7$, $p=0.007$). There was no significant difference between the groups in this respect (session I: low vs. even, SSRI vs. placebo: $t_{64}=1.4$, $p=0.159$; low vs. high, SSRI vs. placebo: $t_{64}=1.0$, $p=0.291$; even vs. high, SSRI vs. placebo: $t_{64}=-0.2$, $p=0.783$; session II: low vs. even, SSRI vs. placebo: $t_{64}=0.06$, $p=0.949$; low vs. high, SSRI vs. placebo: $t_{64}=0.6$, $p=0.491$; even vs. high, SSRI vs. placebo: $t_{64}=0.6$, $p=0.549$). Overall, this demonstrates that subjects learned to dissociate decks, as their willingness to gamble differed depending on each deck's win likelihood as a function of time, and this effect was not modulated by the drug.” (page 5/6, results)

Looking forward to seeing the final version of this paper.

Reviewer #3 (Remarks to the Author):

The authors have thoroughly addressed my previous comments and added several helpful analyses. The manuscript has become clearer and more convincing as a result. I have a few outstanding issues, all of them except the first are very minor:

We thank the reviewer for constructive comments that have enabled improvement to our manuscript. Below we address the reviewer's remaining suggestions.

1) Two reviewers (R2/R3) highlighted the disconnect between the model-free and model-based analyses. The text changes implemented in the manuscript in response to this comment do not sufficiently address this question. Previous outcomes are weighted equally in the logistic regression analysis but are recency-weighted (dependent on the learning rate) in the computational modelling analysis and it is not clear which of the two better explains behaviour. In particular, it is unclear whether the correlation reported in Figure 4 is simply due to similar constructs being measured (in which case the plot is a good sanity check but maybe better placed in the Supplement), or indeed an interesting result that adds further robustness to the conclusions, which is how this result is portrayed at the moment. This can be tested by simulating agents with different positive/negative learning rates (e.g., varying both independently between 0.1-0.8) and establishing the correlations between the two types of analyses in these artificial datasets. I feel this is important to fully establish if the correlation adds value to the manuscript or is a result of the analysis pipeline. I would not insist on this point if the two results in Fig2 and Fig3 were not the core message of this manuscript. It seems crucial that readers can immediately understand whether and how these two types of analyses relate conceptually and methodologically.

We apologise for an insufficient description of the methodological relationship between the two analyses. We deliberately designed the regression (model-agnostic analysis of gambling preference) and RL model (model-based analysis of learning rates) to complement one another, addressing a similar question in two different ways. On this basis, we expected strong correlations between the results to begin with. Arguably, the model-based analysis is more sensitive, at a cost of greater flexibility in fitting the data. The RL model can mimic the regression by fitting the data with very low learning rates, thus weighting outcomes almost equally (while maintaining low choice stochasticity by a compensatory increase in the inverse temperatures). In practice, however, the model fit with medium learning rates (~ 0.1) indicating that subjects did weight recent outcomes more heavily.

To demonstrate the expected correlation between the results of the two analyses, we followed the reviewer's suggestion and simulated artificial data for 5 different models (10 data sets per model), in which we randomly varied positive and negative learning rates independently between 0.1-0.8 for all 66 agents (resulting in substantial variability in learning asymmetries across agents). Next, we ran a logistic regression analysis on the simulated data (equivalent to our analysis of the real data) and averaged the results across the 10 simulated data sets for each model. Subsequently, we computed correlations between an asymmetric effect of cumulative success and failure on gambling (regression) and an asymmetry in learning (computational model), equivalent to Fig. 4 of the original manuscript. Here, we found a highly significant relationship between these two measures across all simulated data sets (ranging between 0.82-0.87, all $p < 2.6e-17$). Note, however, that this means that up to 30% of the variance of the models' predictions do not overlap.

We thank the reviewer for this valuable suggestion and agree it is important to illustrate the relationship between the two measures. Reviewer #1 had initially asked us to move Fig. 4 to the main manuscript and reviewer #2 has now asked us to fortify this analysis further (as described above in #2.3). Considering the results above, and the requests of the two other reviewers, we decided to keep Fig. 4 in the main manuscript instead of moving it back to the supplementary material. We hope the reviewer agrees with our line of reasoning.

“The benefit of having both analyses is that the model-based analysis is more sensitive, albeit at a cost of greater flexibility in fitting the data. Specifically, reinforcement learning

modelling can mimic our regression analysis by fitting the data with very low learning rates, thus weighting outcomes almost equally. However, by fitting the data with higher learning rates, it can also place substantially greater weight on recent outcomes. We additionally illustrate the correspondence between these two measures (regression and reinforcement learning modelling) in simulations with a wide range of parameter settings. Briefly, we simulated artificial data from five models, in which we randomly varied positive and negative learning rates independently across agents. Next, we ran logistic regression analyses on the artificial data and computed correlations between an asymmetric effect of cumulative success and failure on gambling (regression) and an asymmetry in learning (computational model). Here, we found a highly significant relationship across all simulated data sets (r ranging between 0.82-0.87, all $p < 2.6e-17$), providing further evidence for the relationship between these two measures.” (page 10, results)

2) I think other readers will be interested in the additional models that have been tested in response to the reviewer’s comments and included in the rebuttal, but these are currently not part of the revised manuscript. It would be good to mention them at least briefly in the supplement, maybe in the text or figure legends (e.g., related to points 2c and 4: three learning rates for different decks – worse fit; six learning rates - not estimable; deck-unspecific cumulative failure/success – worse fit).

We agree with the reviewer that readers may be interested in these additional models. As suggested, we have now added the results in the revised supplementary table of the model comparison (cf. Supplementary Table S1).

3) Similarly, given the authors have put effort into this analysis, I can see no disadvantage in including the simulation of the optimal learning rate as a supplementary figure (response to previous point 4). It is now mentioned briefly in the legend of Fig3, but readers might wonder how this was established.

We agree with the reviewer and now present this result as a novel supplementary figure S2:

Supplementary Figure S2. Simulations with different learning rates (LRs).

We simulated data for artificial agents ($n=66$, 10 data sets each) with different LRs, ranging from 0.01-0.09 (blue) and 0.1-1 (black), respectively and computed the average net reward gained in the experiment. We also compared this to the net reward gained using LRs derived from fitting to real data (green), as well as to the net reward gained by subjects in the real experiment (red). This analysis revealed that LRs in the range of ≈ 0.1 - 0.7 reap highest reward, with an ‘optimal’ LR in the range of ≈ 0.3 - 0.6 . The results also indicate that lower (e.g., ≤ 0.05), and higher (≥ 0.8) LRs are detrimental to performance. Overall, the mean LR at the population level in our data was 0.125, which allowed subjects, in the real experiment, to reap a net reward that was close to the net reward of artificial agents with an ‘optimal’ LR.

4) Related to my previous point 5 (parameter recovery), I think the confusion was purely down to the x and y-axis labels in Figure S3. It is somewhat confusing to refer to x as ‘fitted to real data’ and y as ‘fitted to simulated data’. I suggest changing these labels to something a bit more intuitive and which conveys both axes relate to the artificial data (e.g., “simulated parameter (chosen based on parameters in real data)” and “recovered parameter” or similar).

We have changed the labels of the axes (x-axis: *simulated*; y-axis: *recovered*) of all panels in the now revised Suppl. Fig. S5 accordingly (cf. Supplementary Material).

5) p-values are still indicated as $p < 0.001$ etc in the results - please report precise p-values.

We apologize for this omission and now report precise p-values throughout the revised manuscript (e.g., page 6, 9, 10 of the revised manuscript).

6) Statements relating to non-significant effects (e.g., Effects were similar across drug groups”) are sometimes framed as if they provide support for the null, but this would require Bayesian statistics (frequentist statistics cannot provide evidence for the null). Bayesian model evidence for the null model (compared to alternatives) are very easy to compute using free software such as JASP. Alternatively, the statements should be reframed as not supporting a difference, rather than as evidence for the null, e.g. “There was no evidence for a difference between drug groups.”

We agree with the reviewer and thus modified the description of non-significant effects accordingly throughout the manuscript, stating there was no evidence for differences between drug groups (e.g., page 6, 7, and 8).

7) In Figure3, in the individual data points, it is noticeable that the between-subject variance is much smaller for negative outcomes in session II both for the SSRI and placebo group. Do the authors have any thoughts on why this might be?

The reviewer is right that between-subject variance, depicted as SEM in the respective figures, is smaller in session II as compared to session I. Importantly, between-subject variance was in fact smaller for negative LR_s in the 2nd session for both the SSRI *and* the placebo group (SSRI, session I: 0.0069, session II: 0.0030; placebo, session I: 0.0071, session II: 0.0026). Moreover, it was reduced to a comparable extent for, e.g., positive LR_s (SSRI, session I: 0.0080, session II: 0.0047; placebo, session I: 0.0065, session II: 0.0055) and for most main results presented in the manuscript, including logistic regression coefficients and model parameter estimates. One explanation might be that subjects were already familiar with the experimental task in the 2nd session, which might have reduced between-subject variance due to less variable gambling behaviour or strategies across the whole population. Critically however, this was the case for all subjects involved in the study. We thus believe it did not impact the main results due to the between-subject pharmacological design of the study.

REVIEWERS' COMMENTS:

Reviewer #2 (Remarks to the Author):

I would like to thank the authors for responding to my recommendations. I hope they will agree that the paper is now much tighter. The only issue which I think has been overlooked is the mention of serotonin not leading to any detrimental or advantageous outcomes in the abstract. I appreciate if the final version of the paper has something along these lines.

I congratulate the authors for this important piece of work.

Reviewer #3 (Remarks to the Author):

The reviewers have done a great job addressing all my comments. I congratulate them on a very nice piece of work!

We thank the referees and editors for the constructive review process, and their positive evaluation of our revised manuscript “Serotonin modulates asymmetric learning from reward and punishment in healthy human volunteers” (COMMSBIO-20-3363B). We have addressed the remaining concern as described below. Additionally, we have modified the title of the manuscript and implemented other changes as suggested by the editors in the editorial requests checklist.

Reviewer #2 (Remarks to the Author):

I would like to thank the authors for responding to my recommendations. I hope they will agree that the paper is now much tighter. The only issue which I think has been overlooked is the mention of serotonin not leading to any detrimental or advantageous outcomes in the abstract. I appreciate if the final version of the paper has something along these lines.

I congratulate the authors for this important piece of work.

We thank the reviewer for the appreciation of our work and their suggestions that helped us to substantially improve the manuscript. We have addressed the remaining concern as described below.

*“This valence-dependent learning asymmetry increases subjects’ tendency to avoid actions as a function of cumulative failure **without leading to detrimental, or advantageous, outcomes.**” (page 1, abstract)*

Reviewer #3 (Remarks to the Author):

The reviewers have done a great job addressing all my comments. I congratulate them on a very nice piece of work!

We are grateful to the reviewer for the constructive review process that led to an improved version of our manuscript and the appreciation of our work.